# Robust Pruning at Initialization

**Soufiane Hayou, Jean-Francois Ton, Arnaud Doucet & Yee Whye Teh**
Department of Statistics
University of Oxford
United Kingdom
`{soufiane.hayou, ton, doucet, teh}@stats.ox.ac.uk`

## Abstract

Overparameterized Neural Networks (NN) display state-of-the-art performance. However, there is a growing need for smaller, energy-efficient, neural networks to be able to use machine learning applications on devices with limited computational resources. A popular approach consists of using pruning techniques. While these techniques have traditionally focused on pruning pre-trained NN (LeCun et al., 1990; Hassibi et al., 1993), recent work by Lee et al. (2018) has shown promising results when pruning at initialization. However, for Deep NNs, such procedures remain unsatisfactory as the resulting pruned networks can be difficult to train and, for instance, they do not prevent one layer from being fully pruned. In this paper, we provide a comprehensive theoretical analysis of Magnitude and Gradient based pruning at initialization and training of sparse architectures. This allows us to propose novel principled approaches which we validate experimentally on a variety of NN architectures.

## 1 Introduction

Overparameterized deep NNs have achieved state of the art (SOTA) performance in many tasks (Nguyen and Hein, 2018; Du et al., 2019; Zhang et al., 2016; Neyshabur et al., 2019). However, it is impractical to implement such models on small devices such as mobile phones. To address this problem, network pruning is widely used to reduce the time and space requirements both at training and test time. The main idea is to identify weights that do not contribute significantly to the model performance based on some criterion, and remove them from the NN. However, most pruning procedures currently available can only be applied after having trained the full NN (LeCun et al., 1990; Hassibi et al., 1993; Mozer and Smolensky, 1989; Dong et al., 2017) although methods that consider pruning the NN during training have become available. For example, Louizos et al. (2018) propose an algorithm which adds a $L_0$ regularization on the weights to enforce sparsity while Carreira-Perpiñán and Idelbayev (2018); Alvarez and Salzmann (2017); Li et al. (2020) propose the inclusion of compression inside training steps. Other pruning variants consider training a secondary network that learns a pruning mask for a given architecture (Li et al. (2020); Liu et al. (2019)).

Recently, Frankle and Carbin (2019) have introduced and validated experimentally the Lottery Ticket Hypothesis which conjectures the existence of a sparse subnetwork that achieves similar performance to the original NN. These empirical findings have motivated the development of pruning at initialization such as SNIP (Lee et al. (2018)) which demonstrated similar performance to classical pruning methods of pruning-after-training. Importantly, pruning at initialization never requires training the complete NN and is thus more memory efficient, allowing to train deep NN using limited computational resources. However, such techniques may suffer from different problems. In particular, nothing prevents such methods from pruning one whole layer of the NN, making it untrainable. More generally, it is typically difficult to train the resulting pruned NN (Li et al., 2018). To solve this situation, Lee et al. (2020) try to tackle this issue by enforcing dynamical isometry using orthogonal weights, while Wang et al. (2020) (GraSP) uses Hessian based pruning to preserve gradient flow. Other work by Tanaka et al. (2020) considers a data-agnostic iterative approach using the concept of synaptic flow in order to avoid the layer-collapse phenomenon (pruning a whole layer). In our work, we use principled scaling and re-parameterization to solve this issue, and show numerically that our algorithm achieves SOTA performance on CIFAR10, CIFAR100, TinyImageNet and ImageNet in some scenarios and remains competitive in others.

Table 1: Classification accuracies on CIFAR10 for Resnet with varying depths and sparsities using SNIP (Lee et al. (2018)) and our algorithm SBP-SR

|  | ALGORITHM | 90% | 95% | 98% | 99.5% | 99.9% |
|---|---|---|---|---|---|---|
| RESNET32 | SNIP | $92.26 \pm 0.32$ | $91.18 \pm 0.17$ | $87.78 \pm 0.16$ | $77.56 \pm 0.36$ | $9.98 \pm 0.08$ |
|  | SBP-SR | $\mathbf{92.56 \pm 0.06}$ | $91.21 \pm 0.30$ | $\mathbf{88.25 \pm 0.35}$ | $\mathbf{79.54 \pm 1.12}$ | $\mathbf{51.56 \pm 1.12}$ |
| RESNET50 | SNIP | $91.95 \pm 0.13$ | $92.12 \pm 0.34$ | $89.26 \pm 0.23$ | $80.49 \pm 2.41$ | $19.98 \pm 14.12$ |
|  | SBP-SR | $\mathbf{92.05 \pm 0.06}$ | $\mathbf{92.74 \pm 0.32}$ | $\mathbf{89.57 \pm 0.21}$ | $\mathbf{82.68 \pm 0.52}$ | $\mathbf{58.76 \pm 1.82}$ |
| RESNET104 | SNIP | $93.25 \pm 0.53$ | $92.98 \pm 0.12$ | $91.58 \pm 0.19$ | $33.63 \pm 33.27$ | $10.11 \pm 0.09$ |
|  | SBP-SR | $\mathbf{94.69 \pm 0.13}$ | $\mathbf{93.88 \pm 0.17}$ | $\mathbf{92.08 \pm 0.14}$ | $\mathbf{87.47 \pm 0.23}$ | $\mathbf{72.70 \pm 0.48}$ |

In this paper, we provide novel algorithms for Sensitivity-Based Pruning (SBP), i.e. pruning schemes that prune a weight $W$ based on the magnitude of $|W \frac{\partial \mathcal{L}}{\partial W}|$ at initialization where $\mathcal{L}$ is the loss. Experimentally, compared to other available one-shot pruning schemes, these algorithms provide state-of the-art results (this might not be true in some regimes). Our work is motivated by a new theoretical analysis of gradient back-propagation relying on the mean-field approximation of deep NN (Hayou et al., 2019; Schoenholz et al., 2017; Poole et al., 2016; Yang and Schoenholz, 2017; Xiao et al., 2018; Lee et al., 2018; Matthews et al., 2018). Our contribution is threefold:

- For deep fully connected FeedForward NN (FFNN) and Convolutional NN (CNN), it has been previously shown that only an initialization on the so-called Edge of Chaos (EOC) make models trainable; see e.g. (Schoenholz et al., 2017; Hayou et al., 2019). For such models, we show that an EOC initialization is also necessary for SBP to be efficient. Outside this regime, one layer can be fully pruned.
- For these models, pruning pushes the NN out of the EOC making the resulting pruned model difficult to train. We introduce a simple rescaling trick to bring the pruned model back in the EOC regime, making the pruned NN easily trainable.
- Unlike FFNN and CNN, we show that Resnets are better suited for pruning at initialization since they 'live' on the EOC by default (Yang and Schoenholz, 2017). However, they can suffer from exploding gradients, which we resolve by introducing a re-parameterization, called 'Stable Resnet' (SR). The performance of the resulting SBP-SR pruning algorithm is illustrated in Table 1: SBP-SR allows for pruning up to 99.5% of ResNet104 on CIFAR10 while still retaining around 87% test accuracy.

The precise statements and proofs of the theoretical results are given in the Supplementary. Appendix H also includes the proof of a weak version of the Lottery Ticket Hypothesis (Frankle and Carbin, 2019) showing that, starting from a randomly initialized NN, there exists a subnetwork initialized on the EOC.

## 2 SENSITIVITY PRUNING FOR FFNN/CNN AND THE RESCALING TRICK

### 2.1 SETUP AND NOTATIONS

Let $x$ be an input in $\mathbb{R}^d$. A NN of depth $L$ is defined by

$$y^l(x) = \mathcal{F}_l(W^l, y^{l-1}(x)) + B^l, \quad 1 \leq l \leq L, \tag{1}$$

where $y^l(x)$ is the vector of pre-activations, $W^l$ and $B^l$ are respectively the weights and bias of the $l^{\text{th}}$ layer and $\mathcal{F}_l$ is a mapping that defines the nature of the layer. The weights and bias are initialized with $W^l \overset{\text{iid}}{\sim} \mathcal{N}(0, \sigma_w^2/v_l)$, where $v_l$ is a scaling factor used to control the variance of $y^l$, and $B^l \overset{\text{iid}}{\sim} \mathcal{N}(0, \sigma_b^2)$. Hereafter, $M_l$ denotes the number of weights in the $l^{th}$ layer, $\phi$ the activation function and $[m : n] := \{m, m+1, ..., n\}$ for $m \leq n$. Two examples of such architectures are:

- **Fully connected FFNN**. For a FFNN of depth $L$ and widths $(N_l)_{0 \leq l \leq L}$, we have $v_l = N_{l-1}$, $M_l = N_{l-1} N_l$ and

$$y_i^1(x) = \sum_{j=1}^{d} W_{ij}^1 x_j + B_i^1, \quad y_i^l(x) = \sum_{j=1}^{N_{l-1}} W_{ij}^l \phi(y_j^{l-1}(x)) + B_i^l \quad \text{for } l \geq 2. \tag{2}$$

- **CNN**. For a 1D CNN of depth $L$, number of channels $(n_l)_{l \leq L}$, and number of neurons per channel $(N_l)_{l \leq L}$, we have

$$y^1_{i,\alpha}(x) = \sum_{j=1}^{n_{l-1}} \sum_{\beta \in ker_l} W^1_{i,j,\beta} x_{j,\alpha+\beta} + b^1_i, \ y^l_{i,\alpha}(x) = \sum_{j=1}^{n_{l-1}} \sum_{\beta \in ker_l} W^l_{i,j,\beta} \phi(y^{l-1}_{j,\alpha+\beta}(x)) + b^l_i, \text{ for } l \geq 2,$$

(3)

where $i \in [1 : n_l]$ is the channel index, $\alpha \in [0 : N_l - 1]$ is the neuron location, $ker_l = [-k_l : k_l]$ is the filter range, and $2k_l + 1$ is the filter size. To simplify the analysis, we assume hereafter that $N_l = N$ and $k_l = k$ for all $l$. Here, we have $v_l = n_{l-1}(2k+1)$ and $M_l = n_{l-1}n_l(2k+1)$. We assume periodic boundary conditions; so $y^l_{i,\alpha} = y^l_{i,\alpha+N} = y^l_{i,\alpha-N}$. Generalization to multidimensional convolutions is straightforward.

When no specific architecture is mentioned, $(W^l_i)_{1 \leq i \leq M_l}$ denotes the weights of the $l^{\text{th}}$ layer. In practice, a pruning algorithm creates a binary mask $\delta$ over the weights to force the pruned weights to be zero. The neural network after pruning is given by

$$y^l(x) = \mathcal{F}_l(\delta^l \circ W^l, y^{l-1}(x)) + B^l,$$

(4)

where $\circ$ is the Hadamard (i.e. element-wise) product. In this paper, we focus on pruning at initialization. The mask is typically created by using a vector $g^l$ of the same dimension as $W^l$ using a mapping of choice (see below), we then prune the network by keeping the weights that correspond to the top $k$ values in the sequence $(g^l_i)_{i,l}$ where $k$ is fixed by the sparsity that we want to achieve. There are three popular types of criteria in the literature :

- **Magnitude based pruning (MBP)**: We prune weights based on the magnitude $|W|$.

- **Sensitivity based pruning (SBP)**: We prune the weights based on the values of $|W \frac{\partial \mathcal{L}}{\partial W}|$ where $\mathcal{L}$ is the loss. This is motivated by $\mathcal{L}_W \approx \mathcal{L}_{W=0} + W \frac{\partial \mathcal{L}}{\partial W}$ used in SNIP (Lee et al. (2018)).

- **Hessian based pruning (HBP)**: We prune the weights based on some function that uses the Hessian of the loss function as in GraSP (Wang et al., 2020).

In the remainder of the paper, we focus exclusively on SBP while our analysis of MBP is given in Appendix E. We leave HBP for future work. However, we include empirical results with GraSP (Wang et al., 2020) in Section 4.

Hereafter, we denote by $s$ the sparsity, i.e. the fraction of weights we want to prune. Let $A_l$ be the set of indices of the weights in the $l^{\text{th}}$ layer that are pruned, i.e. $A_l = \{i \in [1 : M_l], \text{ s.t. } \delta^l_i = 0\}$. We define the critical sparsity $s_{cr}$ by

$$s_{cr} = \min\{s \in (0,1), \text{ s.t. } \exists l, |A_l| = M_l\},$$

where $|A_l|$ is the cardinality of $A_l$. Intuitively, $s_{cr}$ represents the maximal sparsity we are allowed to choose without fully pruning at least one layer. $s_{cr}$ is random as the weights are initialized randomly. Thus, we study the behaviour of the expected value $\mathbb{E}[s_{cr}]$ where, hereafter, **all expectations are taken w.r.t. to the random initial weights**. This provides theoretical guidelines for pruning at initialization.

For all $l \in [1 : L]$, we define $\alpha_l$ by $v_l = \alpha_l N$ where $N > 0$, and $\zeta_l > 0$ such that $M_l = \zeta_l N^2$, where we recall that $v_l$ is a scaling factor controlling the variance of $y^l$ and $M_l$ is the number of weights in the $l^{\text{th}}$ layer. This notation assumes that, in each layer, the number of weights is quadratic in the number of neurons, which is satisfied by classical FFNN and CNN architectures.

## 2.2 SENSITIVITY-BASED PRUNING (SBP)

SBP is a data-dependent pruning method that uses the data to compute the gradient *with* backpropagation at initialization (one-shot pruning). We randomly sample a batch and compute the gradients of the loss with respect to each weight. The mask is then defined by $\delta^l_i = \mathbb{I}(|W^l_i \frac{\partial \mathcal{L}}{\partial W^l_i}| \geq t_s)$, where $t_s = |W \frac{\partial \mathcal{L}}{\partial W}|^{(k_s)}$ and $k_s = (1-s) \sum_l M_l$ and $|W \frac{\partial \mathcal{L}}{\partial W}|^{(k_s)}$ is the $k_s^{\text{th}}$ order statistics of the sequence $(|W^l_i \frac{\partial \mathcal{L}}{\partial W^l_i}|)_{1 \leq l \leq L, 1 \leq i \leq M_l}$.

However, this simple approach suffers from the well-known exploding/vanishing gradients problem which renders the first/last few layers respectively susceptible to be completely pruned. We give a formal definition to this problem.

**Definition 1** (Well-conditioned & ill-conditioned NN). *Let $m_l = \mathbb{E}[|W_1^l \frac{\partial \mathcal{L}}{\partial W_1^l}|^2]$ for $l \in [1 : L]$. We say that the NN is well-conditioned if there exist $A, B > 0$ such that for all $L \geq 1$ and $l \in [1 : L]$ we have $A \leq m_l/m_L \leq B$, and it is ill-conditioned otherwise.*

Understanding the behaviour of gradients at initialization is thus crucial for SBP to be efficient. Using a mean-field approach, such analysis has been carried out in (Schoenholz et al., 2017; Hayou et al., 2019; Xiao et al., 2018; Poole et al., 2016; Yang, 2019) where it has been shown that an initialization known as the EOC is beneficial for DNN training. The mean-field analysis of DNNs relies on two standard approximations that we will also use here.

**Approximation 1** (Mean-Field Approximation). *When $N_l \gg 1$ for FFNN or $n_l \gg 1$ for CNN, we use the approximation of infinitely wide NN. This means infinite number of neurons per layer for fully connected layers and infinite number of channels per layer for convolutional layers.*

**Approximation 2** (Gradient Independence). *The weights used for forward propagation are independent from those used for back-propagation.*

These two approximations are ubiquitous in literature on the mean-field analysis of neural networks. They have been used to derive theoretical results on signal propagation (Schoenholz et al., 2017; Hayou et al., 2019; Poole et al., 2016; Yang, 2019; Yang and Schoenholz, 2017; Yang et al., 2019) and are also key tools in the derivation of the Neural Tangent Kernel (Jacot et al., 2018; Arora et al., 2019; Hayou et al., 2020). Approximation 1 simplifies the analysis of the forward propagation as it allows the derivation of closed-form formulas for covariance propagation. Approximation 2 does the same for back-propagation. See Appendix A for a detailed discussion of these approximations. Throughout the paper, we provide numerical results that substantiate the theoretical results that we derive using these two approximations. We show that these approximations lead to excellent match between theoretical results and numerical experiments.

**Edge of Chaos (EOC):** For inputs $x, x'$, let $c^l(x, x')$ be the correlation between $y^l(x)$ and $y^l(x')$. From (Schoenholz et al., 2017; Hayou et al., 2019), there exists a so-called correlation function $f$ that depends on $(\sigma_w, \sigma_b)$ such that $c^{l+1}(x, x') = f(c^l(x, x'))$. Let $\chi(\sigma_b, \sigma_w) = f'(1)$. The EOC is the set of hyperparameters $(\sigma_w, \sigma_b)$ satisfying $\chi(\sigma_b, \sigma_w) = 1$. When $\chi(\sigma_b, \sigma_w) > 1$, we are in the Chaotic phase, the gradient explodes and $c^l(x, x')$ converges exponentially to some $c < 1$ for $x \neq x'$ and the resulting output function is discontinuous everywhere. When $\chi(\sigma_b, \sigma_w) < 1$, we are in the Ordered phase where $c^l(x, x')$ converges exponentially fast to 1 and the NN outputs constant functions. Initialization on the EOC allows for better information propagation (see Supplementary for more details).

Hence, by leveraging the above results, we show that an initialization outside the EOC will lead to an ill-conditioned NN.

**Theorem 1** (EOC Initialization is crucial for SBP). *Consider a NN of type (2) or (3) (FFNN or CNN). Assume $(\sigma_w, \sigma_b)$ are chosen on the ordered phase, i.e. $\chi(\sigma_b, \sigma_w) < 1$, then the NN is ill-conditioned. Moreover, we have*

$$\mathbb{E}[s_{cr}] \leq \frac{1}{L}\left(1 + \frac{\log(\kappa L N^2)}{\kappa}\right) + \mathcal{O}\left(\frac{1}{\kappa^2 \sqrt{L N^2}}\right),$$

*where $\kappa = |\log \chi(\sigma_b, \sigma_w)|/8$. If $(\sigma_w, \sigma_b)$ are on the EOC, i.e. $\chi(\sigma_b, \sigma_w) = 1$, then the NN is well-conditioned. In this case, $\kappa = 0$ and the above upper bound no longer holds.*

The proof of Theorem 1 relies on the behaviour of the gradient norm at initialization. On the ordered phase, the gradient norm vanishes exponentially quickly as it back-propagates, thus resulting in an ill-conditioned network. We use another approximation for the sake of simplification of the proof (Approximation 3 in the Supplementary) but the result holds without this approximation although the resulting constants would be a bit different. Theorem 1 shows that the upper bound decreases the farther $\chi(\sigma_b, \sigma_w)$ is from 1, i.e. the farther the initialization is from the EOC. For constant width FFNN with $L = 100$, $N = 100$ and $\kappa = 0.2$, the theoretical upper bound is $\mathbb{E}[s_{cr}] \lesssim 27\%$ while we obtain $\mathbb{E}[s_{cr}] \approx 22\%$ based on 10 simulations. A similar result can be obtained when the NN is initialized on the chaotic phase; in this case too, the NN is ill-conditioned. To illustrate these results, Figure 1 shows the impact of the initialization with sparsity $s = 70\%$. The dark area in Figure 1(b) corresponds to layers that are fully pruned in the chaotic phase due to exploding gradients. Using an EOC initialization, Figure 1(a) shows that pruned weights are well distributed in the NN, ensuring that no layer is fully pruned.

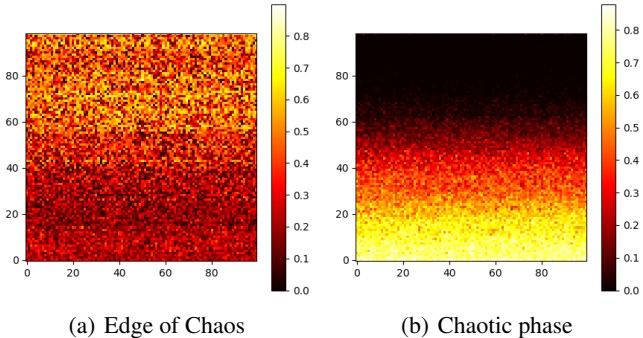

(a) Edge of Chaos      (b) Chaotic phase

Figure 1: Percentage of weights kept after SBP applied to a randomly initialized FFNN with depth 100 and width 100 for 70% sparsity on MNIST. Each pixel $(i, j)$ corresponds to a neuron and shows the proportion of connections to neuron $(i, j)$ that have not been pruned. The EOC (a) allows us to preserve a uniform spread of the weights, whereas the Chaotic phase (b), due to exploding gradients, prunes entire layers.

## 2.3 Training Pruned Networks Using the Rescaling Trick

We have shown previously that an initialization on the EOC is crucial for SBP. However, we have not yet addressed the key problem of training the resulting pruned NN. This can be very challenging in practice (Li et al., 2018), especially for deep NN.

Consider as an example a FFNN architecture. After pruning, we have for an input $x$

$$\hat{y}_i^l(x) = \sum_{j=1}^{N_{l-1}} W_{ij}^l \delta_{ij}^l \phi(\hat{y}_j^{l-1}(x)) + B_i^l, \quad \text{for } l \geq 2, \tag{5}$$

where $\delta$ is the pruning mask. While the original NN initialized on the EOC was satisfying $c^{l+1}(x, x') = f(c^l(x, x'))$ for $f'(1) = \chi(\sigma_b, \sigma_w) = 1$, the pruned architecture leads to $\hat{c}^{l+1}(x, x') = f_{\text{pruned}}(\hat{c}^l(x, x'))$ with $f'_{\text{pruned}}(1) \neq 1$, hence *pruning destroys the EOC*. Consequently, the pruned NN will be difficult to train (Schoenholz et al., 2017; Hayou et al., 2019) especially if it is deep. Hence, we propose to bring the pruned NN back on the EOC. This approach consists of rescaling the weights obtained after SBP in each layer by factors that depend on the pruned architecture itself.

**Proposition 1** (Rescaling Trick). *Consider a NN of type (2) or (3) (FFNN or CNN) initialized on the EOC. Then, after pruning, the pruned NN is not initialized on the EOC anymore. However, the rescaled pruned NN*

$$y^l(x) = \mathcal{F}(\rho^l \circ \delta^l \circ W^l, y^{l-1}(x)) + B^l, \quad \text{for } l \geq 1, \tag{6}$$

*where*

$$\rho_{ij}^l = (\mathbb{E}[N_{l-1}(W_{i1}^l)^2 \delta_{i1}^l])^{-\frac{1}{2}} \text{ for FFNN}, \quad \rho_{i,j,\beta}^l = (\mathbb{E}[n_{l-1}(W_{i,1,\beta}^l)^2 \delta_{i,1,\beta}^l])^{-\frac{1}{2}} \text{ for CNN}, \tag{7}$$

*is initialized on the EOC. (The scaling is constant across $j$).*

The scaling factors in equation 7 are easily approximated using the weights kept after pruning. Algorithm 1 (see Appendix I) details a practical implementation of this rescaling technique for FFNN. We illustrate experimentally the benefits of this approach in Section 4.

## 3 Sensitivity-Based Pruning for Stable Residual Networks

Resnets and their variants (He et al., 2015; Huang et al., 2017) are currently the best performing models on various classification tasks (CIFAR10, CIFAR100, ImageNet etc (Kolesnikov et al., 2019)). Thus, understanding Resnet pruning at initialization is of crucial interest. Yang and Schoenholz (2017) showed that Resnets naturally 'live' on the EOC. Using this result, we show that Resnets are actually better suited to SBP than FFNN and CNN. However, Resnets suffer from an exploding gradient problem (Yang and Schoenholz, 2017) which might affect the performance of SBP. We

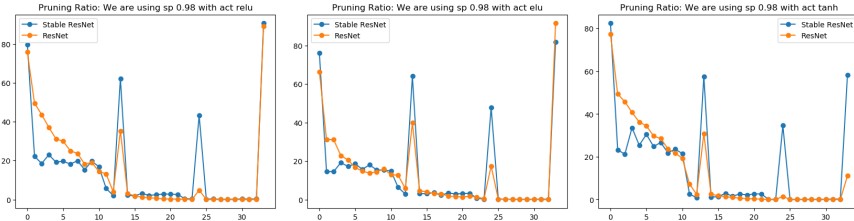

Figure 2: Percentage of non-pruned weights per layer in a ResNet32 for our Stable ResNet32 and standard Resnet32 with Kaiming initialization on CIFAR10. With Stable Resnet, we prune less aggressively weights in the deeper layers than for standard Resnet.

address this issue by introducing a new Resnet parameterization. Let a standard Resnet architecture be given by

$$y^1(x) = \mathcal{F}(W^1, x), \quad y^l(x) = y^{l-1}(x) + \mathcal{F}(W^l, y^{l-1}), \quad \text{for } l \geq 2, \tag{8}$$

where $\mathcal{F}$ defines the blocks of the Resnet. Hereafter, we assume that $\mathcal{F}$ is either of the form (2) or (3) (FFNN or CNN).

The next theorem shows that Resnets are well-conditioned independently from the initialization and are thus well suited for pruning at initialization.

**Theorem 2** (Resnet are Well-Conditioned). *Consider a Resnet with either Fully Connected or Convolutional layers and ReLU activation function. Then for all $\sigma_w > 0$, the Resnet is well-conditioned. Moreover, for all $l \in \{1, ..., L\}$, we have $m^l = \Theta((1 + \frac{\sigma_w^2}{2})^L)$.*

The above theorem proves that Resnets are always well-conditioned. However, taking a closer look at $m^l$, which represents the variance of the pruning criterion (Definition 1), we see that it grows exponentially in the number of layers $L$. Therefore, this could lead to a 'higher variance of pruned networks' and hence high variance test accuracy. To this end, we propose a Resnet parameterization which we call Stable Resnet. Stable Resnets prevent the second moment from growing exponentially as shown below.

**Proposition 2** (Stable Resnet). *Consider the following Resnet parameterization*

$$y^l(x) = y^{l-1}(x) + \frac{1}{\sqrt{L}}\mathcal{F}(W^l, y^{l-1}), \quad for \, l \geq 2, \tag{9}$$

*then the NN is well-conditioned for all $\sigma_w > 0$. Moreover, for all $l \leq L$ we have $m^l = \Theta(L^{-1})$.*

In Proposition 2, $L$ is not the number of layers but the number of blocks. For example, ResNet32 has 15 blocks and 32 layers, hence $L = 15$. Figure 2 shows the percentage of weights in each layer kept after pruning ResNet32 and Stable ResNet32 at initialization. The jumps correspond to limits between sections in ResNet32 and are caused by max-pooling. Within each section, Stable Resnet tends to have a more uniform distribution of percentages of weights kept after pruning compared to standard Resnet. In Section 4 we show that this leads to better performance of Stable Resnet compared to standard Resnet. Further theoretical and experimental results for Stable Resnets are presented in (Hayou et al., 2021).

In the next proposition, we establish that, unlike FFNN or CNN, we do not need to rescale the pruned Resnet for it to be trainable as it lives naturally on the EOC before and after pruning.

**Proposition 3** (Resnet live on the EOC even after pruning). *Consider a Residual NN with blocks of type FFNN or CNN. Then, after pruning, the pruned Residual NN is initialized on the EOC.*

## 4 EXPERIMENTS

In this section, we illustrate empirically the theoretical results obtained in the previous sections. We validate the results on MNIST, CIFAR10, CIFAR100 and Tiny ImageNet.

### 4.1 INITIALIZATION AND RESCALING

According to Theorem 1, an EOC initialization is necessary for the network to be well-conditioned. We train FFNN with tanh activation on MNIST, varying depth $L \in \{2, 20, 40, 60, 80, 100\}$ and

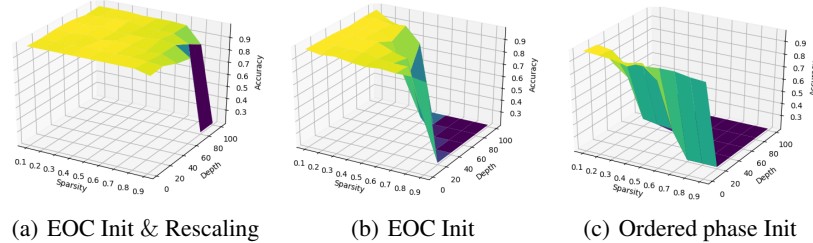

|           (a) EOC Init & Rescaling          |          (b) EOC Init          |       (c) Ordered phase Init       |

Figure 3: Accuracy on MNIST with different initialization schemes including EOC with rescaling, EOC without rescaling, Ordered phase, with varying depth and sparsity. This shows that rescaling to be on the EOC allows us to train not only much deeper but also sparser models.

sparsity $s \in \{10\%, 20\%, .., 90\%\}$. We use SGD with batchsize 100 and learning rate $10^{-3}$, which we found to be optimal using a grid search with an exponential scale of 10. Figure 3 shows the test accuracy after 10k iterations for 3 different initialization schemes: *Rescaled EOC*, *EOC*, *Ordered*. On the Ordered phase, the model is untrainable when we choose sparsity $s > 40\%$ and depth $L > 60$ as one layer being fully pruned. For an EOC initialization, the set $(s, L)$ for which NN are trainable becomes larger. However, the model is still untrainable for highly sparse deep networks as the sparse NN is no longer initialized on the EOC (see Proposition 1). As predicted by Proposition 1, after application of the rescaling trick to bring back the pruned NN on the EOC, the pruned NN can be trained appropriately.

Table 2: Classification accuracies for CIFAR10 and CIFAR100 after pruning

|  | CIFAR10 | | | CIFAR100 | | |
|---|---|---|---|---|---|---|
| SPARSITY | 90% | 95% | 98% | 90% | 95% | 98% |
| **ResNet32** (NO PRUNING) | 94.80 | - | - | 74.64 | - | - |
| OBD LECUN ET AL. (1990) | 93.74 | 93.58 | 93.49 | 73.83 | 71.98 | 67.79 |
| RANDOM PRUNING | 89.95±0.23 | 89.68±0.15 | 86.13±0.25 | 63.13±2.94 | 64.55±0.32 | 19.83±3.21 |
| MBP | 90.21±0.55 | 88.35±0.75 | 86.83±0.27 | 67.07±0.31 | 64.92±0.77 | 59.53±2.19 |
| SNIP LEE ET AL. (2018) | 92.26 ± 0.32 | 91.18 ± 0.17 | 87.78 ± 0.16 | 69.31 ± 0.52 | 65.63 ± 0.15 | 55.70 ± 1.13 |
| GRASP WANG ET AL. (2020) | 92.20±0.31 | 91.39±0.25 | **88.70±0.42** | 69.24 ± 0.24 | 66.50 ± 0.11 | 58.43 ± 0.43 |
| GRASP-SR | 92.30±0.19 | 91.16±0.13 | 87.8 ± 0.32 | 69.12 ± 0.15 | 65.49 ± 0.21 | 58.63 ± 0.23 |
| SYNFLOW TANAKA ET AL. (2020) | 92.01±0.22 | **91.67±0.17** | 88.10 ± 0.25 | 69.03 ± 0.20 | 65.23 ± 0.31 | 58.73 ± 0.30 |
| SBP-SR (STABLE RESNET) | **92.56 ± 0.06** | 91.21 ± 0.30 | 88.25 ± 0.35 | **69.51 ± 0.21** | **66.72 ± 0.12** | **59.51 ± 0.15** |
| **ResNet50** (NO PRUNING) | 94.90 | - | - | 74.9 | - | - |
| RANDOM PRUNING | 85.11±4.51 | 88.76±0.21 | 85.32±0.47 | 65.67±0.57 | 60.23±2.21 | 28.32±10.35 |
| MBP | 90.11 ± 0.32 | 89.06 ± 0.09 | 87.32 ± 0.16 | 68.51 ± 0.21 | 63.32 ± 1.32 | 55.21 ± 0.35 |
| SNIP | 91.95 ± 0.13 | 92.12 ± 0.34 | 89.26 ± 0.23 | 70.43 ± 0.43 | 67.85 ± 1.02 | 60.38 ± 0.78 |
| GRASP | **92.10 ± 0.21** | 91.74 ± 0.35 | **89.97± 0.25** | 70.53±0.32 | 67.84±0.25 | 63.88±0.45 |
| SYNFLOW | 92.05 ± 0.20 | 91.83 ± 0.23 | 89.61 ± 0.17 | 70.43±0.30 | 67.95±0.22 | 63.95±0.11 |
| SBP-SR | 92.05 ± 0.06 | **92.74± 0.32** | 89.57 ± 0.21 | **71.79 ± 0.13** | **68.98 ± 0.15** | **64.45 ± 0.34** |
| **ResNet104** (NO PRUNING) | 94.92 | - | - | 75.24 | - | - |
| RANDOM PRUNING | 89.80±0.33 | 87.86±1.22 | 85.52±2.12 | 66.73±1.32 | 64.98±0.11 | 30.31±4.51 |
| MBP | 90.05 ± 1.23 | 88.95±0.65 | 87.83±1.21 | 69.57±0.35 | 64.31±0.78 | 60.21±2.41 |
| SNIP | 93.25 ± 0.53 | 92.98 ± 0.12 | 91.58 ± 0.19 | 71.94 ± 0.22 | 68.73±0.09 | 63.31 ± 0.41 |
| GRASP | 93.08 ± 0.17 | 92.93 ± 0.09 | 91.19±0.35 | 73.33±0.21 | 70.95 ± 1.12 | 66.91±0.33 |
| SYNFLOW | 93.43 ± 0.10 | 92.85 ± 0.18 | 91.03±0.25 | 72.85±0.20 | 70.33 ± 0.15 | 67.02±0.10 |
| SBP-SR | **94.69 ± 0.13** | **93.88 ± 0.17** | **92.08 ± 0.14** | **74.17 ± 0.11** | **71.84 ± 0.13** | **67.73 ± 0.28** |

## 4.2 RESNET AND STABLE RESNET

Although Resnets are adapted to SBP (i.e. they are always well-conditioned for all $\sigma_w > 0$), Theorem 2 shows that the magnitude of the pruning criterion grows exponentially w.r.t. the depth $L$. To resolve this problem we introduced Stable Resnet. We call our pruning algorithm for ResNet SBP-SR (SBP with Stable Resnet). Theoretically, we expect SBP-SR to perform better than other methods for deep Resnets according to Proposition 2. Table 2 shows test accuracies for ResNet32, ResNet50 and ResNet104 with varying sparsities $s \in \{90\%, 95\%, 98\%\}$ on CIFAR10 and CIFAR100. For all our experiments, we use a setup similar to (Wang et al., 2020), i.e. we use SGD for 160 and 250 epochs for CIFAR10 and CIFAR100, respectively. We use an initial learning rate of 0.1 and decay it by 0.1

at $1/2$ and $3/4$ of the number of total epoch. In addition, we run all our experiments 3 times to obtain more stable and reliable test accuracies. As in (Wang et al., 2020), we adopt Resnet architectures where we doubled the number of filters in each convolutional layer. As a baseline, we include pruning results with the classical OBD pruning algorithm (LeCun et al., 1990) for ResNet32 (train $\rightarrow$ prune $\rightarrow$ repeat). We compare our results against other algorithms that prune at initialization, such as SNIP (Lee et al., 2018), which is a SBP algorithm, GraSP (Wang et al., 2020) which is a Hessian based pruning algorithm, and SynFlow (Tanaka et al., 2020), which is an iterative data-agnostic pruning algorithm. As we increase the depth, SBP-SR starts to outperform other algorithms that prune at initialization (SBP-SR outperforms all other algorithms with ResNet104 on CIFAR10 and CIFAR100). Furthermore, using GraSP on Stable Resnet did not improve the result of GraSP on standard Resnet, as our proposed Stable Resnet analysis only applies to gradient based pruning. The analysis of Hessian based pruning could lead to similar techniques for improving trainability, which we leave for future work.

Table 3: Classification accuracies on Tiny ImageNet for Resnet with varying depths

|  | ALGORITHM | 85% | 90% | 95% |
|---|---|---|---|---|
| RESNET32 | SBP-SR | **57.25 ± 0.09** | **55.67 ± 0.21** | 50.63±0.21 |
|  | SNIP | 56.92± 0.33 | 54.99±0.37 | 49.48±0.48 |
|  | GRASP | **57.25±0.11** | 55.53±0.11 | 51.34±0.29 |
|  | SYNFLOW | 56.75±0.09 | 55.60±0.07 | **51.50±0.21** |
| RESNET50 | SBP-SR | **59.8±0.18** | **57.74±0.06** | **53.97±0.27** |
|  | SNIP | 58.91±0.23 | 56.15±0.31 | 51.19±0.47 |
|  | GRASP | 58.46±0.29 | 57.48±0.35 | 52.5±0.41 |
|  | SYNFLOW | 59.31±0.17 | **57.67±0.15** | 53.14±0.31 |
| RESNET104 | SBP-SR | **62.84±0.13** | **61.96±0.11** | **57.9±0.31** |
|  | SNIP | 59.94±0.34 | 58.14±0.28 | 54.9±0.42 |
|  | GRASP | 61.1±0.41 | 60.14±0.38 | 56.36±0.51 |
|  | SYNFLOW | 61.71±0.08 | 60.81±0.14 | 55.91±0.43 |

To confirm these results, we also test SBP-SR against other pruning algorithms on Tiny ImageNet. We train the models for 300 training epochs to make sure all algorithms converge. Table 3 shows test accuracies for SBP-SR, SNIP, GraSP, and SynFlow for $s \in \{85\%, 90\%, 95\%\}$. Although SynFlow competes or outperforms GraSP in many cases, SBP-SR has a clear advantage over SynFlow and other algorithms, especially for deep networks as illustrated on ResNet104.

Additional results with ImageNet dataset are provided in Appendix F.

## 4.3 ReScaling Trick and CNNs

The theoretical analysis of Section 2 is valid for Vanilla CNN i.e. CNN without pooling layers. With pooling layers, the theory of signal propagation applies to sections between successive pooling layers; each of those section can be seen as Vanilla CNN. This applies to standard CNN architectures such as VGG. As a *toy example*, we show in Table 4 the test accuracy of a pruned V-CNN with sparsity $s = 50\%$ on MNIST dataset. Similar to FFNN results in Figure 3, the combination of the EOC Init and the ReScaling trick allows for pruning deep V-CNN (depth 100) while ensuring their trainability.

Table 4: Test accuracy on MNIST with V-CNN for different depths with sparsity 50% using SBP(SNIP)

|  | $L = 10$ | $L = 50$ | $L = 100$ |
|---|---|---|---|
| ORDERED PHASE INIT | 98.12±0.13 | 10.00±0.0 | 10.00±0.0 |
| EOC INIT | 98.20±0.17 | 98.75±0.11 | 10.00±0.0 |
| EOC + RESCALING | 98.18±0.21 | 98.90±0.07 | 99.15±0.08 |

However, V-CNN is a toy example that is generally not used in practice. Standard CNN architectures such as VGG are popular among practitioners since they achieve SOTA accuracy on many tasks. Table 5 shows test accuracies for SNIP, SynFlow, and our EOC+ReScaling trick for VGG16 on CIFAR10. Our results are close to the results presented by Frankle et al. (2020). These three

algorithms perform similarly. From a theoretical point of view, our ReScaling trick applies to vanilla CNNs without pooling layers, hence, adding pooling layers might cause a deterioration. However, we know that the signal propagation theory applies to vanilla blocks inside VGG (i.e. the sequence of convolutional layers between two successive pooling layers). The larger those vanilla blocks are, the better our ReScaling trick performs. We leverage this observation by training a modified version of VGG, called 3xVGG16, which has the same number of pooling layers as VGG16, and 3 times the number of convolutional layers inside each vanilla block. Numerical results in Table 5 show that the EOC initialization with the ReScaling trick outperforms other algorithms, which confirms our hypothesis. However, the architecture 3xVGG16 is not a standard architecture and it does not seem to improve much the test accuracy of VGG16. An adaptation of the ReScaling trick to standard VGG architectures would be of great value and is left for future work.

Table 5: Classification accuracy on CIFAR10 for VGG16 and 3xVGG16 with varying sparsities

|  | ALGORITHM | 85% | 90% | 95% |
|---|---|---|---|---|
| VGG16 | SNIP | 93.09±0.11 | 92.97±0.08 | 92.61±0.10 |
|  | SYNFLOW | 93.21±0.13 | 93.05±0.11 | 92.19±0.12 |
|  | EOC + RESCALING | 93.15±0.12 | 92.90±0.15 | 92.70±0.06 |
| 3xVGG16 | SNIP | 93.30±0.10 | 93.12±0.20 | 92.85±0.15 |
|  | SYNFLOW | 92.95±0.13 | 92.91±0.21 | 92.70±0.20 |
|  | EOC + RESCALING | **93.97±0.17** | **93.75±0.15** | **93.40±0.16** |

**Summary of numerical results.** We summarize in Table 6 our numerical results. The letter 'C' refers to 'Competition' between algorithms in that setting, and indicates no clear winner is found, while the dash means no experiment has been run with this setting. We observe that our algorithm SBP-SR consistently outperforms other algorithms in a variety of settings.

Table 6: Which algorithm performs better? (according to our results)

| DATASET | ARCHITECTURE | 85% | 90% | 95% | 98% |
|---|---|---|---|---|---|
| CIFAR10 | RESNET32 | - | C | C | GRASP |
|  | RESNET50 | - | C | SBP-SR | GRASP |
|  | RESNET104 | - | SBP-SR | SBP-SR | SBP-SR |
|  | VGG16 | C | C | C | - |
|  | 3xVGG16 | EOC+RESC | EOC+RESC | EOC+RESC | - |
| CIFAR100 | RESNET32 | - | SBP-SR | SBP-SR | SBP-SR |
|  | RESNET50 | - | SBP-SR | SBP-SR | SBP-SR |
|  | RESNET104 | - | SBP-SR | SBP-SR | SBP-SR |
| TINY IMAGENET | RESNET32 | C | C | SYNFLOW | - |
|  | RESNET50 | SBP-SR | C | SBP-SR | - |
|  | RESNET104 | SBP-SR | SBP-SR | SBP-SR | - |

## 5 CONCLUSION

In this paper, we have formulated principled guidelines for SBP at initialization. For FNNN and CNN, we have shown that an initialization on the EOC is necessary followed by the application of a simple rescaling trick to train the pruned network. For Resnets, the situation is markedly different. There is no need for a specific initialization but Resnets in their original form suffer from an exploding gradient problem. We propose an alternative Resnet parameterization called Stable Resnet, which allows for more stable pruning. Our theoretical results have been validated by extensive experiments on MNIST, CIFAR10, CIFAR100, Tiny ImageNet and ImageNet. Compared to other available one-shot pruning algorithms, we achieve state-of the-art results in many scenarios.

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

# A   DISCUSSION ABOUT APPROXIMATIONS 1 AND 2

## A.1   APPROXIMATION 1: INFINITE WIDTH APPROXIMATION

**FeedForward Neural Network**

Consider a randomly initialized FFNN of depth $L$, widths $(N_l)_{1 \leq l \leq L}$, weights $W_{ij}^l \overset{iid}{\sim} \mathcal{N}(0, \frac{\sigma_w^2}{N_{l-1}})$ and bias $B_i^l \overset{iid}{\sim} \mathcal{N}(0, \sigma_b^2)$, where $\mathcal{N}(\mu, \sigma^2)$ denotes the normal distribution of mean $\mu$ and variance $\sigma^2$. For some input $x \in \mathbb{R}^d$, the propagation of this input through the network is given by

$$y_i^1(x) = \sum_{j=1}^d W_{ij}^1 x_j + B_i^1, \tag{10}$$

$$y_i^l(x) = \sum_{j=1}^{N_{l-1}} W_{ij}^l \phi(y_j^{l-1}(x)) + B_i^l, \quad \text{for } l \geq 2. \tag{11}$$

Where $\phi : \mathbb{R} \to \mathbb{R}$ is the activation function. When we take the limit $N_{l-1} \to \infty$, the Central Limit Theorem implies that $y_i^l(x)$ is a Gaussian variable for any input $x$. This approximation by infinite width solution results in an error of order $\mathcal{O}(1/\sqrt{N_{l-1}})$ (standard Monte Carlo error). More generally, an approximation of the random process $y_i^l(.)$ by a Gaussian process was first proposed by Neal (1995) in the single layer case and has been recently extended to the multiple layer case by Lee et al. (2018) and Matthews et al. (2018). We recall here the expressions of the limiting Gaussian process kernels. For any input $x \in \mathbb{R}^d$, $\mathbb{E}[y_i^l(x)] = 0$ so that for any inputs $x, x' \in \mathbb{R}^d$

$$\begin{aligned}
\kappa^l(x, x') &= \mathbb{E}[y_i^l(x) y_i^l(x')] \\
&= \sigma_b^2 + \sigma_w^2 \mathbb{E}[\phi(y_i^{l-1}(x))\phi(y_i^{l-1}(x'))] \\
&= \sigma_b^2 + \sigma_w^2 F_\phi(\kappa^{l-1}(x, x), \kappa^{l-1}(x, x'), \kappa^{l-1}(x', x')),
\end{aligned}$$

where $F_\phi$ is a function that only depends on $\phi$. This provides a simple recursive formula for the computation of the kernel $\kappa^l$; see, e.g., Lee et al. (2018) for more details.

**Convolutional Neural Networks**

Similar to the FFNN case, the infinite width approximation with 1D CNN (introduced in the main paper) yields a recursion for the kernel. However, the infinite width here means infinite number of channels, and results in an error $\mathcal{O}(1/\sqrt{n_{l-1}})$. The kernel in this case depends on the choice of the neurons in the channel and is given by

$$\kappa_{\alpha,\alpha'}^l(x, x') = \mathbb{E}[y_{i,\alpha}^l(x) y_{i,\alpha'}^l(x')] = \sigma_b^2 + \frac{\sigma_w^2}{2k+1} \sum_{\beta \in ker} \mathbb{E}[\phi(y_{1,\alpha+\beta}^{l-1}(x))\phi(y_{1,\alpha'+\beta}^{l-1}(x'))]$$

so that

$$\kappa_{\alpha,\alpha'}^l(x, x') = \sigma_b^2 + \frac{\sigma_w^2}{2k+1} \sum_{\beta \in ker} F_\phi(\kappa_{\alpha+\beta,\alpha'+\beta}^{l-1}(x, x), \kappa_{\alpha+\beta,\alpha'+\beta}^{l-1}(x, x'), \kappa_{\alpha+\beta,\alpha'+\beta}^{l-1}(x', x')).$$

The convolutional kernel $\kappa_{\alpha,\alpha'}^l$ has the 'self-averaging' property; i.e. it is an average over the kernels corresponding to different combination of neurons in the previous layer. However, it is easy to simplify the analysis in this case by studying the average kernel per channel defined by $\hat{\kappa}^l = \frac{1}{N^2} \sum_{\alpha,\alpha'} \kappa_{\alpha,\alpha'}^l$. Indeed, by summing terms in the previous equation and using the fact that we use circular padding, we obtain

$$\hat{\kappa}^l(x, x') = \sigma_b^2 + \sigma_w^2 \frac{1}{N^2} \sum_{\alpha,\alpha'} F_\phi(\kappa_{\alpha,\alpha'}^{l-1}(x, x), \kappa_{\alpha,\alpha'}^{l-1}(x, x'), \kappa_{\alpha,\alpha'}^{l-1}(x', x')).$$

This expression is similar in nature to that of FFNN. We will use this observation in the proofs.

Note that our analysis only requires the approximation that, in the infinite width limit, for any two inputs $x, x'$, the variables $y_i^l(x)$ and $y_i^l(x')$ are Gaussian with covariance $\kappa^l(x, x')$ for FFNN, and

$y_{i,\alpha}^l(x)$ and $y_{i,\alpha'}^l(x')$ are Gaussian with covariance $\kappa_{\alpha,\alpha'}^l(x,x')$ for CNN. We do not need the much stronger approximation that the process $y_i^l(x)$ ($y_{i,\alpha}^l(x)$ for CNN) is a Gaussian process.

**Residual Neural Networks**

The infinite width limit approximation for ResNet yields similar results with an additional residual terms. It is straighforward to see that, in the case a ResNet with FFNN-type layers, we have that

$$\kappa^l(x,x') = \kappa^{l-1}(x,x') + \sigma_b^2 + \sigma_w^2 F_\phi(\kappa^{l-1}(x,x), \kappa^{l-1}(x,x'), \kappa^{l-1}(x',x')),$$

whereas for ResNet with CNN-type layers, we have that

$$\kappa_{\alpha,\alpha'}^l(x,x') = \kappa_{\alpha,\alpha'}^{l-1}(x,x') + \sigma_b^2$$
$$+ \frac{\sigma_w^2}{2k+1} \sum_{\beta \in ker} F_\phi(\kappa_{\alpha+\beta,\alpha'+\beta}^{l-1}(x,x), \kappa_{\alpha+\beta,\alpha'+\beta}^{l-1}(x,x'), \kappa_{\alpha+\beta,\alpha'+\beta}^{l-1}(x',x')).$$

### A.2 APPROXIMATION 2: GRADIENT INDEPENDENCE

For gradient back-propagation, an essential assumption in prior literature in Mean-Field analysis of DNNs is that of the gradient independence which is similar in nature to the practice of feedback alignment (Lillicrap et al., 2016). This approximation allows for derivation of recursive formulas for gradient back-propagation, and it has been extensively used in literature and verified empirically; see references below.

**Gradient Covariance back-propagation:** this approximation was used to derive analytical formulas for gradient covariance back-propagation in (Hayou et al., 2019; Schoenholz et al., 2017; Yang and Schoenholz, 2017; Lee et al., 2018; Poole et al., 2016; Xiao et al., 2018; Yang, 2019). It was shown empirically through simulations that it is an excellent approximation for FFNN in Schoenholz et al. (2017), for Resnets in Yang and Schoenholz (2017) and for CNN in Xiao et al. (2018).

**Neural Tangent Kernel (NTK):** this approximation was implicitly used by Jacot et al. (2018) to derive the recursive formula of the infinite width Neural Tangent Kernel (See Jacot et al. (2018), Appendix A.1). Authors have found that this approximation yields excellent match with exact NTK. It was also exploited later in (Arora et al., 2019; Hayou et al., 2020) to derive the infinite NTK for different architectures. The difference between the infinite width NTK $\Theta$ and the empirical (exact) NTK $\hat{\Theta}$ was studied in Lee et al. (2019) where authors have shown that $\|\Theta - \hat{\Theta}\|_F = \mathcal{O}(N^{-1})$ where $N$ is the width of the NN.

More precisely, we use the approximation that, for wide neural networks, the weights used for forward propagation are independent from those used for back-propagation. When used for the computation of gradient covariance and Neural Tangent Kernel, this approximation was proven to give the exact computation for standard architectures such as FFNN, CNN and ResNets, without BatchNorm in Yang (2019) (section D.5). Even with BatchNorm, in Yang et al. (2019), authors have found that the Gradient Independence approximation matches empirical results.

This approximation can be alternatively formulated as an assumption instead of an approximation as in Yang and Schoenholz (2017).

**Assumption 1 (Gradient Independence):** The gradients are computed using an i.i.d. version of the weights used for forward propagation.

## B PRELIMINARY RESULTS

Let $x$ be an input in $\mathbb{R}^d$. In its general form, a neural network of depth $L$ is given by the following set of forward propagation equations

$$y^l(x) = \mathcal{F}_l(W^l, y^{l-1}(x)) + B^l, \quad 1 \le l \le L, \tag{12}$$

where $y^l(x)$ is the vector of pre-activations and $W^l$ and $B^l$ are respectively the weights and bias of the $l^{th}$ layer. $\mathcal{F}_l$ is a mapping that defines the nature of the layer. The weights and bias are initialized with $W^l \overset{iid}{\sim} \mathcal{N}(0, \frac{\sigma_w^2}{v_l})$ where $v_l$ is a scaling factor used to control the variance of $y^l$, and $B^l \overset{iid}{\sim} \mathcal{N}(0, \sigma_b^2)$. Hereafter, we denote by $M_l$ the number of weights in the $l^{th}$ layer, $\phi$ the activation function and $[n:m]$ the set of integers $\{n, n+1, ..., m\}$ for $n \le m$. Two examples of such architectures are:

- **Fully-connected FeedForward Neural Network (FFNN)**
  For a fully connected feedforward neural network of depth $L$ and widths $(N_l)_{0 \leq l \leq L}$, the forward propagation of the input through the network is given by

$$y_i^1(x) = \sum_{j=1}^{d} W_{ij}^1 x_j + B_i^1,$$

$$y_i^l(x) = \sum_{j=1}^{N_{l-1}} W_{ij}^l \phi(y_j^{l-1}(x)) + B_i^l, \quad \text{for } l \geq 2. \tag{13}$$

  Here, we have $v_l = N_{l-1}$ and $M_l = N_{l-1}N_l$.

- **Convolutional Neural Network (CNN/ConvNet)**
  For a 1D convolutional neural network of depth $L$, number of channels $(n_l)_{l \leq L}$ and number of neurons per channel $(N_l)_{l \leq L}$. we have

$$y_{i,\alpha}^1(x) = \sum_{j=1}^{n_{l-1}} \sum_{\beta \in ker_l} W_{i,j,\beta}^1 x_{j,\alpha+\beta} + b_i^1,$$

$$y_{i,\alpha}^l(x) = \sum_{j=1}^{n_{l-1}} \sum_{\beta \in ker_l} W_{i,j,\beta}^l \phi(y_{j,\alpha+\beta}^{l-1}(x)) + b_i^l, \quad \text{for } l \geq 2, \tag{14}$$

  where $i \in [1 : n_l]$ is the channel index, $\alpha \in [0 : N_l - 1]$ is the neuron location, $ker_l = [-k_l : k_l]$ is the filter range and $2k_l + 1$ is the filter size. To simplify the analysis, we assume hereafter that $N_l = N$ and $k_l = k$ for all $l$. Here, we have $v_l = n_{l-1}(2k + 1)$ and $M_l = n_{l-1}n_l(2k + 1)$. We assume periodic boundary conditions, so $y_{i,\alpha}^l = y_{i,\alpha+N}^l = y_{i,\alpha-N}^l$. Generalization to multidimensional convolutions is straighforward.

**Notation:** Hereafter, for FFNN layers, we denote by $q^l(x)$ the variance of $y_1^l(x)$ (the choice of the index 1 is not crucial since, by the mean-field approximation, the random variables $(y_i^l(x))_{i \in [1:N_l]}$ are iid Gaussian variables). We denote by $q^l(x, x')$ the covariance between $y_1^l(x)$ and $y_1^l(x')$, and $c_1^l(x, x')$ the corresponding correlation. For gradient back-propagation, for some loss function $\mathcal{L}$, we denote by $\tilde{q}^l(x, x')$ the gradient covariance defined by $\tilde{q}^l(x, x') = \mathbb{E}\left[\frac{\partial \mathcal{L}}{\partial y_1^l}(x)\frac{\partial \mathcal{L}}{\partial y_1^l}(x')\right]$. Similarly, $\tilde{q}^l(x)$ denotes the gradient variance at point $x$.

For CNN layers, we use similar notation accross channels. More precisely, we denote by $q_\alpha^l(x)$ the variance of $y_{1,\alpha}^l(x)$ (the choice of the index 1 is not crucial here either since, by the mean-field approximation, the random variables $(y_{i,\alpha}^l(x))_{i \in [1:N_l]}$ are iid Gaussian variables). We denote by $q_{\alpha,\alpha'}^l(x, x')$ the covariance between $y_{1,\alpha}^l(x)$ and $y_{1,\alpha'}^l(x')$, and $c_{\alpha,\alpha'}^l(x, x')$ the corresponding correlation.

As in the FFNN case, we define the gradient covariance by $\tilde{q}_{\alpha,\alpha'}^l(x, x') = \mathbb{E}\left[\frac{\partial \mathcal{L}}{\partial y_{1,\alpha}^l}(x)\frac{\partial \mathcal{L}}{\partial y_{1,\alpha'}^l}(x')\right]$.

### B.1   WARMUP : SOME RESULTS FROM THE MEAN-FIELD THEORY OF DNNS

We start by recalling some results from the mean-field theory of deep NNs.

#### B.1.1   COVARIANCE PROPAGATION

**Covariance propagation for FFNN:**
In Section A.1, we presented the recursive formula for covariance propagation in a FFNN, which we derive using the Central Limit Theorem. More precisely, for two inputs $x, x' \in \mathbb{R}^d$, we have

$$q^l(x, x') = \sigma_b^2 + \sigma_w^2 \mathbb{E}[\phi(y_i^{l-1}(x))\phi(y_i^{l-1}(x'))].$$

This can be rewritten as

$$q^l(x, x') = \sigma_b^2 + \sigma_w^2 \mathbb{E}\left[\phi\left(\sqrt{q^l(x)}Z_1\right)\phi\left(\sqrt{q^l(x')}(c^{l-1}Z_1 + \sqrt{1 - (c^{l-1})^2}Z_2)\right)\right],$$

where $c^{l-1} := c^{l-1}(x, x')$.

With a ReLU activation function, we have

$$q^l(x, x') = \sigma_b^2 + \frac{\sigma_w^2}{2}\sqrt{q^l(x)}\sqrt{q^l(x')}f(c^{l-1}),$$

where $f$ is the ReLU correlation function given by (Hayou et al. (2019))

$$f(c) = \frac{1}{\pi}(c \arcsin c + \sqrt{1-c^2}) + \frac{1}{2}c.$$

**Covariance propagation for CNN:**

Similar to the FFNN case, it is straightforward to derive recusive formula for the covariance. However, in this case, the independence is across channels and not neurons. Simple calculus yields

$$q_{\alpha,\alpha'}^l(x, x') = \mathbb{E}[y_{i,\alpha}^l(x)y_{i,\alpha'}^l(x')] = \sigma_b^2 + \frac{\sigma_w^2}{2k+1}\sum_{\beta \in ker}\mathbb{E}[\phi(y_{1,\alpha+\beta}^{l-1}(x))\phi(y_{1,\alpha'+\beta}^{l-1}(x'))]$$

Using a ReLU activation function, this becomes

$$q_{\alpha,\alpha'}^l(x, x') = \sigma_b^2 + \frac{\sigma_w^2}{2k+1}\sum_{\beta \in ker}\sqrt{q_{\alpha+\beta}^l(x)}\sqrt{q_{\alpha'+\beta}^l(x')}f(c_{\alpha+\beta,\alpha'+\beta}^{l-1}(x, x')).$$

**Covariance propagation for ResNet with ReLU :**

This case is similar to the non residual case. However, an added residual term shows up in the recursive formula. For ResNet with FFNN layers, we have

$$q^l(x, x') = q^{l-1}(x, x') + \sigma_b^2 + \frac{\sigma_w^2}{2}\sqrt{q^l(x)}\sqrt{q^l(x')}f(c^{l-1})$$

and for ResNet with CNN layers, we have

$$q_{\alpha,\alpha'}^l(x, x') = q_{\alpha,\alpha'}^{l-1}(x, x') + \sigma_b^2 + \frac{\sigma_w^2}{2k+1}\sum_{\beta \in ker}\sqrt{q_{\alpha+\beta}^l(x)}\sqrt{q_{\alpha'+\beta}^l(x')}f(c_{\alpha+\beta,\alpha'+\beta}^{l-1}(x, x')).$$

### B.1.2 GRADIENT COVARIANCE BACK-PROPAGATION

**Gradiant Covariance back-propagation for FFNN:**

Let $\mathcal{L}$ be the loss function. Let $x$ be an input. The back-propagation of the gradient is given by the set of equations

$$\frac{\partial \mathcal{L}}{\partial y_i^l} = \phi'(y_i^l)\sum_{j=1}^{N_{l+1}}\frac{\partial \mathcal{L}}{\partial y_j^{l+1}}W_{ji}^{l+1}.$$

Using the approximation that the weights used for forward propagation are independent from those used in backpropagation, we have as in Schoenholz et al. (2017)

$$\tilde{q}^l(x) = \tilde{q}^{l+1}(x)\frac{N_{l+1}}{N_l}\chi(q^l(x)),$$

where $\chi(q^l(x)) = \sigma_w^2\mathbb{E}[\phi(\sqrt{q^l(x)}Z)^2]$.

**Gradient Covariance back-propagation for CNN:**

Similar to the FFNN case, we have that

$$\frac{\partial \mathcal{L}}{\partial W_{i,j,\beta}^l} = \sum_{\alpha}\frac{\partial \mathcal{L}}{\partial y_{i,\alpha}^l}\phi(y_{j,\alpha+\beta}^{l-1})$$

and

$$\frac{\partial \mathcal{L}}{\partial y_{i,\alpha}^l} = \sum_{j=1}^n\sum_{\beta \in ker}\frac{\partial \mathcal{L}}{\partial y_{j,\alpha-\beta}^{l+1}}W_{i,j,\beta}^{l+1}\phi'(y_{i,\alpha}^l).$$

Using the approximation of Gradient independence and averaging over the number of channels (using CLT) we have that

$$\mathbb{E}[\frac{\partial \mathcal{L}}{\partial y_{i,\alpha}^l}^2] = \frac{\sigma_w^2 \mathbb{E}[\phi'(\sqrt{q_\alpha^l(x)}Z)^2]}{2k+1} \sum_{\beta \in ker} \mathbb{E}[\frac{\partial \mathcal{L}}{\partial y_{i,\alpha-\beta}^{l+1}}^2].$$

We can get similar recursion to that of the FFNN case by summing over $\alpha$ and using the periodic boundary condition, this yields

$$\sum_\alpha \mathbb{E}[\frac{\partial \mathcal{L}}{\partial y_{i,\alpha}^l}^2] = \chi(q_\alpha^l(x)) \sum_\alpha \mathbb{E}[\frac{\partial \mathcal{L}}{\partial y_{i,\alpha}^{l+1}}^2].$$

### B.1.3   EDGE OF CHAOS (EOC)

Let $x \in \mathbb{R}^d$ be an input. The convergence of $q^l(x)$ as $l$ increases has been studied by Schoenholz et al. (2017) and Hayou et al. (2019). In particular, under weak regularity conditions, it is proven that $q^l(x)$ converges to a point $q(\sigma_b, \sigma_w) > 0$ independent of $x$ as $l \to \infty$. The asymptotic behaviour of the correlations $c^l(x, x')$ between $y^l(x)$ and $y^l(x')$ for any two inputs $x$ and $x'$ is also driven by $(\sigma_b, \sigma_w)$: the dynamics of $c^l$ is controlled by a function $f$ i.e. $c^{l+1} = f(c^l)$ called the correlation function. The authors define the EOC as the set of parameters $(\sigma_b, \sigma_w)$ such that $\sigma_w^2 \mathbb{E}[\phi'(\sqrt{q(\sigma_b, \sigma_w)}Z)^2] = 1$ where $Z \sim \mathcal{N}(0, 1)$. Similarly the Ordered, resp. Chaotic, phase is defined by $\sigma_w^2 \mathbb{E}[\phi'(\sqrt{q(\sigma_b, \sigma_w)}Z)^2] < 1$, resp. $\sigma_w^2 \mathbb{E}[\phi'(\sqrt{q(\sigma_b, \sigma_w)}Z)^2] > 1$. On the Ordered phase, the gradient will vanish as it backpropagates through the network, and the correlation $c^l(x, x')$ converges exponentially to 1. Hence the output function becomes constant (hence the name 'Ordered phase'). On the Chaotic phase, the gradient explodes and the correlation converges exponentially to some limiting value $c < 1$ which results in the output function being discontinuous everywhere (hence the 'Chaotic' phase name). On the EOC, the second moment of the gradient remains constant throughout the backpropagation and the correlation converges to 1 at a sub-exponential rate, which allows deeper information propagation. Hereafter, $f$ **will always refer to the correlation function**.

### B.1.4   SOME RESULTS FROM THE MEAN-FIELD THEORY OF DEEP FFNNS

Let $\epsilon \in (0, 1)$ and $B_\epsilon = \{(x, x')\mathbb{R}^d : c^1(x, x') < 1 - \epsilon\}$ (For now $B_\epsilon$ is defined only for FFNN).

Using Approximation 1, the following results have been derived by Schoenholz et al. (2017) and Hayou et al. (2019):

- There exist $q, \lambda > 0$ such that $\sup_{x \in \mathbb{R}^d} |q^l - q| \le e^{-\lambda l}$.
- On the Ordered phase, there exists $\gamma > 0$ such that $\sup_{x,x' \in \mathbb{R}^d} |c^l(x, x') - 1| \le e^{-\gamma l}$.
- On the Chaotic phase, For all $\epsilon \in (0, 1)$ there exist $\gamma > 0$ and $c < 1$ such that $\sup_{(x,x') \in B_\epsilon} |c^l(x, x') - c| \le e^{-\gamma l}$.
- For ReLU network on the EOC, we have

$$f(x) \underset{x \to 1-}{=} x + \frac{2\sqrt{2}}{3\pi}(1 - x)^{3/2} + O((1 - x)^{5/2}).$$

- In general, we have

$$f(x) = \frac{\sigma_b^2 + \sigma_w^2 \mathbb{E}[\phi(\sqrt{q}Z_1)\phi(\sqrt{q}Z(x))]}{q}, \tag{15}$$

  where $Z(x) = xZ_1 + \sqrt{1 - x^2}Z_2$ and $Z_1, Z_2$ are iid standard Gaussian variables.
- On the EOC, we have $f'(1) = 1$
- On the Ordered, resp. Chaotic, phase we have that $f'(1) < 1$, resp. $f'(1) > 1$.
- For non-linear activation functions, $f$ is strictly convex and $f(1) = 1$.
- $f$ is increasing on $[-1, 1]$.

- On the Ordered phase and EOC, $f$ has one fixed point which is 1. On the chaotic phase, $f$ has two fixed points: 1 which is unstable, and $c \in (0,1)$ which is a stable fixed point.

- On the Ordered/Chaotic phase, the correlation between gradients computed with different inputs converges exponentially to 0 as we back-progapagate the gradients.

Similar results exist for CNN. Xiao et al. (2018) show that, similarly to the FFNN case, there exists $q$ such that $q_\alpha^l(x)$ converges exponentially to $q$ for all $x, \alpha$, and studied the limiting behaviour of correlation between neurons at the same channel $c_{\alpha,\alpha'}^l(x,x)$ (same input $x$). These correlations describe how features are correlated for the same input. However, they do not capture the behaviour of these features for different inputs (i.e. $c_{\alpha,\alpha'}^l(x,x')$ where $x \neq x'$). We establish this result in the next section.

## B.2 CORRELATION BEHAVIOUR IN CNN IN THE LIMIT OF LARGE DEPTH

**Appendix Lemma 1** (Asymptotic behaviour of the correlation in CNN with smooth activation functions). *We consider a 1D CNN. Let $(\sigma_b, \sigma_w) \in (\mathbb{R}^+)^2$ and $x \neq x'$ be two inputs $\in \mathbb{R}^d$. If $(\sigma_b, \sigma_w)$ are either on the Ordered or Chaotic phase, then there exists $\beta > 0$ such that*

$$\sup_{\alpha,\alpha'} |c_{\alpha,\alpha'}^l(x,x') - c| = \mathcal{O}(e^{-\beta l}),$$

*where $c = 1$ if $(\sigma_b, \sigma_w)$ is in the Ordered phase, and $c \in (0,1)$ if $(\sigma_b, \sigma_w)$ is in the Chaotic phase.*

*Proof.* Let $x \neq x'$ be two inputs and $\alpha, \alpha'$ two nodes in the same channel $i$. From Section B.1, we have that

$$q_{\alpha,\alpha'}^l(x,x') = \mathbb{E}[y_{i,\alpha}^l(x)y_{i,\alpha'}^l(x')] = \frac{\sigma_w^2}{2k+1} \sum_{\beta \in ker} \mathbb{E}[\phi(y_{1,\alpha+\beta}^{l-1}(x))\phi(y_{1,\alpha'+\beta}^{l-1}(x'))] + \sigma_b^2.$$

This yields

$$c_{\alpha,\alpha'}^l(x,x') = \frac{1}{2k+1} \sum_{\beta \in ker} f(c_{\alpha+\beta,\alpha'+\beta}^{l-1}(x,x')),$$

where $f$ is the correlation function.
We prove the result in the Ordered phase, the proof in the Chaotic phase is similar. Let $(\sigma_b, \sigma_w)$ be in the Ordered phase and $c_m^l = \min_{\alpha,\alpha'} c_{\alpha,a'}^l(x,x')$. Using the fact that $f$ is non-decreasing (section B.1), we have that $c_{\alpha,\alpha'}^l(x,x') \geq \frac{1}{2k+1} \sum_{\beta \in ker} c_{\alpha+\beta,\alpha'+\beta}^{l-1}(x,x') \geq f(c_m^{l-1})$. Taking the min again over $\alpha, \alpha'$, we have $c_m^l \geq f(c_m^{l-1})$, therefore $c_m^l$ is non-decreasing and converges to a stable fixed point of $f$. By the convexity of $f$, the limit is 1 (in the Chaotic phase, $f$ has two fixed point, a stable point $c_1 < 1$ and $c_2 = 1$ unstable). Moreover, the convergence is exponential using the fact that $0 < f'(1) < 1$. We conclude using the fact that $\sup_{\alpha,\alpha'} |c_{\alpha,\alpha'}^l(x,x') - 1| = 1 - c_m^l$. $\square$

## C PROOFS FOR SECTION 2 : SBP FOR FFNN/CNN AND THE RESCALING TRICK

In this section, we prove Theorem 1 and Proposition 1. Before proving Theorem 1, we state the degeneracy approximation.

**Approximation 3** (Degeneracy on the Ordered phase). *On the Ordered phase, the correlation $c^l$ and the variance $q^l$ converge exponentially quickly to their limiting values 1 and $q$ respectively. The degeneracy approximation for FFNN states that*

- $\forall x \neq x', c^l(x,x') \approx 1$

- $\forall x, q^l(x) \approx q$

*For CNN,*

- $\forall x \neq x', \alpha, \alpha', c_{\alpha,\alpha'}^l(x,x') \approx 1$

- $\forall x, q_\alpha^l(x) \approx q$

The degeneracy approximation is essential in the proof of Theorem 1 as it allows us to avoid many unnecessary complications. However, the results holds without this approximation although the constants may be a bit different.

**Theorem 1** (Initialization is crucial for SBP). *We consider a FFNN (2) or a CNN (3). Assume $(\sigma_w, \sigma_b)$ are chosen on the ordered, i.e. $\chi(\sigma_b, \sigma_w) < 1$, then the NN is ill-conditioned. Moreover, we have*

$$\mathbb{E}[s_{cr}] \leq \frac{1}{L}\left(1 + \frac{\log(\kappa L N^2)}{\kappa}\right) + \mathcal{O}\left(\frac{1}{\kappa^2 \sqrt{L N^2}}\right),$$

*where $\kappa = |\log \chi(\sigma_b, \sigma_w)|/8$. If $(\sigma_w, \sigma_b)$ are on the EOC, i.e. $\chi(\sigma_b, \sigma_w) = 1$, then the NN is well-conditioned. In this case, $\kappa = 0$ and the above upper bound no longer holds.*

*Proof.* We prove the result using Approximation 3.

1. **Case 1 : Fully connected Feedforward Neural Networks**
   To simplify the notation, we assume that $N_l = N$ and $M_l = N^2$ (i.e. $\alpha_l = 1$ and $\zeta_l = 1$) for all $l$. We prove the result for the Ordered phase, the proof for the Chaotic phase is similar. Let $L_0 \gg 1$, $\epsilon \in (0, 1 - \frac{1}{L_0})$, $L \geq L_0$ and $x \in (\frac{1}{L} + \epsilon, 1)$. With sparsity $x$, we keep $k_x = \lfloor (1-x)LN^2 \rfloor$ weights. We have

   $$\mathbb{P}(s_{cr} \leq x) \geq \mathbb{P}(\max_{i,j} |W_{ij}^1| \left|\frac{\partial \mathcal{L}}{\partial W_{ij}^1}\right| < t^{(k_x)})$$

   where $t^{(k_x)}$ is the $k_x^{th}$ order statistic of the sequence $\{|W_{ij}^l| \left|\frac{\partial \mathcal{L}}{\partial W_{ij}^l}\right|, l > 0, (i,j) \in [1:N]^2\}$.

   We have

   $$\frac{\partial \mathcal{L}}{\partial W_{ij}^l} = \frac{1}{|\mathcal{D}|} \sum_{x \in \mathcal{D}} \frac{\partial \mathcal{L}}{\partial y_i^l(x)} \frac{\partial y_i^l(x)}{\partial W_{ij}^l}$$
   $$= \frac{1}{|\mathcal{D}|} \sum_{x \in \mathcal{D}} \frac{\partial \mathcal{L}}{\partial y_i^l(x)} \phi(y_j^{l-1}(x)).$$

   On the Ordered phase, the variance $q^l(x)$ and the correlation $c^l(x, x')$ converge exponentially to their limiting values $q$, 1 (Section B.1). Under the degeneracy Approximation 3, we have

   - $\forall x \neq x', c^l(x, x') \approx 1$
   - $\forall x, q^l(x) \approx q$

   Let $\tilde{q}^l(x) = \mathbb{E}[\frac{\partial \mathcal{L}}{\partial y_i^l(x)}^2]$ (the choice of $i$ is not important since $(y_i^l(x))_i$ are iid ). Using these approximations, we have that $y_i^l(x) = y_i^l(x')$ almost surely for all $x, x'$. Thus

   $$\mathbb{E}\left[\frac{\partial \mathcal{L}}{\partial W_{ij}^l}^2\right] = \mathbb{E}[\phi(\sqrt{q}Z)^2]\tilde{q}^l(x),$$

   where $x$ is an input. The choice of $x$ is not important in our approximation.
   From Section B.1.2, we have

   $$\tilde{q}_x^l = \tilde{q}_x^{l+1} \frac{N_{l+1}}{N_l}\chi.$$

   Then we obtain

   $$\tilde{q}_x^l = \frac{N_L}{N_l}\tilde{q}_x^L \chi^{L-l} = \tilde{q}_x^L \chi^{L-l},$$

   where $\chi = \sigma_w^2 \mathbb{E}[\phi(\sqrt{q}Z)^2]$ as we have assumed $N_l = N$. Using this result, we have

   $$\mathbb{E}\left[\frac{\partial \mathcal{L}}{\partial W_{ij}^l}^2\right] = A\,\chi^{L-l},$$

where $A = \mathbb{E}[\phi(\sqrt{q}Z)^2]\tilde{q}_x^L$ for an input $x$. Recall that by definition, one has $\chi < 1$ on the Ordered phase.

In the general case, i.e. without the degeneracy approximation on $c^l$ and $q^l$, we can prove that

$$\mathbb{E}\big[\frac{\partial \mathcal{L}}{\partial W_{ij}^l}^2\big] = \Theta(\chi^{L-l})$$

which suffices for the rest of the proof. However, the proof of this result requires many unnecessary complications that do not add any intuitive value to the proof.

In the general case where the widths are different, $\tilde{q}^l$ will also scale as $\chi^{L-l}$ up to a different constant.
Now we want to lower bound the probability

$$\mathbb{P}(\max_{i,j} |W_{ij}^1| \big| \frac{\partial \mathcal{L}}{\partial W_{ij}^1} \big| < t^{(k_x)}).$$

Let $t_\epsilon^{(k_x)}$ be the $k_x^{\text{th}}$ order statistic of the sequence $\{|W_{ij}^l| \big| \frac{\partial \mathcal{L}}{\partial W_{ij}^l} \big|, l > 1 + \epsilon L, (i,j) \in [1 : N]^2\}$. It is clear that $t^{(k_x)} > t_\epsilon^{(k_x)}$, therefore

$$\mathbb{P}(\max_{i,j} |W_{ij}^1| \big| \frac{\partial \mathcal{L}}{\partial W_{ij}^1} \big| < t^{(k_x)}) \geq \mathbb{P}(\max_{i,j} |W_{ij}^1| \big| \frac{\partial \mathcal{L}}{\partial W_{ij}^1} \big| < t_\epsilon^{(k_x)}).$$

Using Markov's inequality, we have that

$$\mathbb{P}(\big| \frac{\partial \mathcal{L}}{\partial W_{ij}^1} \big| \geq \alpha) \leq \frac{\mathbb{E}\big[\big| \frac{\partial \mathcal{L}}{\partial W_{ij}^1} \big|^2\big]}{\alpha^2}. \tag{16}$$

Note that $\text{Var}(\chi^{\frac{l-L}{2}} \big| \frac{\partial \mathcal{L}}{\partial W_{ij}^l} \big|) = A$. In general, the random variables $\chi^{\frac{l-L}{2}} \big| \frac{\partial \mathcal{L}}{\partial W_{ij}^l} \big|$ have a density $f_{ij}^l$ for all $l > 1 + \epsilon L, (i,j) \in [1 : N]^2$, such that $f_{ij}^l(0) \neq 0$. Therefore, there exists a constant $\lambda$ such that for $x$ small enough,

$$\mathbb{P}(\chi^{\frac{l-L}{2}} \big| \frac{\partial \mathcal{L}}{\partial W_{ij}^l} \big| \geq x) \geq 1 - \lambda x.$$

By selecting $x = \chi^{\frac{(1-\epsilon/2)L-1}{2}}$, we obtain

$$\chi^{\frac{l-L}{2}} \times x \leq \chi^{\frac{(1+\epsilon L)-L}{2}} \chi^{\frac{(1-\epsilon/2)L-1}{2}} = \chi^{\epsilon L/2}.$$

Therefore, for $L$ large enough, and all $l > 1 + \epsilon L$, $(i,j) \in [1 : N_l] \times [1 : N_{l-1}]$, we have

$$\mathbb{P}(\big| \frac{\partial \mathcal{L}}{\partial W_{ij}^l} \big| \geq \chi^{\frac{(1-\epsilon/2)L-1}{2}}) \geq 1 - \lambda \chi^{\frac{l-(\epsilon L/2+1)}{2}} \geq 1 - \lambda \chi^{\epsilon L/2}.$$

Now choosing $\alpha = \chi^{\frac{(1-\epsilon/4)L-1}{2}}$ in inequality (16) yields

$$\mathbb{P}(\big| \frac{\partial \mathcal{L}}{\partial W_{ij}^1} \big| \geq \chi^{\frac{(1-\epsilon/4)L-1}{2}}) \geq 1 - A \chi^{\epsilon L/4}.$$

Since we do not know the exact distribution of the gradients, the trick is to bound them using the previous concentration inequalities. We define the event $B := \{\forall (i,j) \in [1 : N] \times [1 : d], \big| \frac{\partial \mathcal{L}}{\partial W_{ij}^1} \big| \leq \chi^{\frac{(1-\epsilon/4)L-1}{2}}\} \cap \{\forall l > 1 + \epsilon L, (i,j) \in [1 : N]^2, \big| \frac{\partial \mathcal{L}}{\partial W_{ij}^l} \big| \geq \chi^{\frac{(1-\epsilon/2)L-1}{2}}\}$.

We have

$$\mathbb{P}(\max_{i,j} |W_{ij}^1| \big| \frac{\partial \mathcal{L}}{\partial W_{ij}^1} \big| < t_\epsilon^{(k_x)}) \geq \mathbb{P}(\max_{i,j} |W_{ij}^1| \big| \frac{\partial \mathcal{L}}{\partial W_{ij}^1} \big| < t_\epsilon^{(k_x)} \big| B)\mathbb{P}(B).$$

But, by conditioning on the event $B$, we also have

$$\mathbb{P}(\max_{i,j}|W_{ij}^1||\frac{\partial\mathcal{L}}{\partial W_{ij}^1}| < t_\epsilon^{(k_x)}|B) \geq \mathbb{P}(\max_{i,j}|W_{ij}^1| < \chi^{-\epsilon L/8}t_\epsilon'^{(k_x)}),$$

where $t_\epsilon'^{(k_x)}$ is the $k_x^{\text{th}}$ order statistic of the sequence $\{|W_{ij}^l|, l > 1 + \epsilon L, (i,j) \in [1:N]^2\}$.

Now, as in the proof of Proposition 4 in Appendix E (MBP section), define $x_{\zeta,\gamma_L} = \min\{y \in (0,1) : \forall x > y, \gamma_L Q_x > Q_{1-(1-x)^{\gamma_L^{2-\zeta}}}\}$, where $\gamma_L = \chi^{-\epsilon L/8}$. Since $\lim_{\zeta\to 2} x_{\zeta,\gamma_L} = 0$, then there exists $\zeta_\epsilon < 2$ such that $x_{\zeta_\epsilon,\gamma_L} = \epsilon + \frac{1}{L}$.

As $L$ grows, $t_\epsilon'^{(k_x)}$ converges to the quantile of order $\frac{x-\epsilon}{1-\epsilon}$. Therefore,

$$\mathbb{P}(\max_{i,j}|W_{ij}^1| < \chi^{-\epsilon L/8}t_\epsilon'^{(k_x)}) \geq \mathbb{P}(\max_{i,j}|W_{ij}^1| < Q_{1-(1-\frac{x-\epsilon}{1-\epsilon})^{\gamma_L^{2-\zeta_\epsilon}}}) + \mathcal{O}(\frac{1}{\sqrt{LN^2}})$$
$$\geq 1 - N^2(\frac{x-\epsilon}{1-\epsilon})^{\gamma_L^{2-\zeta_\epsilon}} + \mathcal{O}(\frac{1}{\sqrt{LN^2}}).$$

Using the above concentration inequalities on the gradient, we obtain

$$\mathbb{P}(B) \geq (1 - A\,\chi^{\epsilon L/4})^{N^2}(1 - \lambda\,\chi^{\epsilon L/2})^{LN^2}.$$

Therefore there exists a constant $\eta > 0$ independent of $\epsilon$ such that

$$\mathbb{P}(B) \geq 1 - \eta LN^2\chi^{\epsilon L/4}.$$

Hence, we obtain

$$\mathbb{P}(s_{cr} \geq x) \leq N^2(\frac{x-\epsilon}{1-\epsilon})^{\gamma_L^{2-\zeta_\epsilon}} + \eta LN^2\chi^{\epsilon L/4} + \mathcal{O}(\frac{1}{\sqrt{LN^2}}).$$

Integration of the previous inequality yields

$$\mathbb{E}[s_{cr}] \leq \epsilon + \frac{1}{L} + \frac{N^2}{1 + \gamma_L^{2-\zeta_\epsilon}} + \eta LN^2\chi^{\epsilon L/4} + \mathcal{O}(\frac{1}{\sqrt{LN^2}}).$$

Now let $\kappa = \frac{|\log(\chi)|}{8}$ and set $\epsilon = \frac{\log(\kappa LN^2)}{\kappa L}$. By the definition of $x_{\zeta_\epsilon}$, we have

$$\gamma_L Q_{x_{\zeta_\epsilon,\gamma_L}} = Q_{1-(1-x_{\zeta_\epsilon,\gamma_L})^{\gamma_L^{2-\zeta_\epsilon}}}.$$

For the left hand side, we have

$$\gamma_L Q_{x_{\zeta_\epsilon,\gamma_L}} \sim \alpha\gamma_L\frac{\log(\kappa LN^2)}{\kappa L}$$

where $\alpha > 0$ is the derivative at 0 of the function $x \to Q_x$. Since $\gamma_L = \kappa LN^2$, we have

$$\gamma_L Q_{x_{\zeta_\epsilon,\gamma_L}} \sim \alpha N^2\log(\kappa LN^2)$$

Which diverges as $L$ goes to infinity. In particular this proves that the right hand side diverges and therefore we have that $(1 - x_{\zeta_\epsilon,\gamma_L})^{\gamma_L^{2-\zeta_\epsilon}}$ converges to 0 as $L$ goes to infinity. Using the asymptotic equivalent of the right hand side as $L \to \infty$, we have $Q_{1-(1-x_{\zeta_\epsilon,\gamma_L})^{\gamma_L^{2-\zeta_\epsilon}}} \sim \sqrt{-2\log((1-x_{\zeta_\epsilon,\gamma_L})^{\gamma_L^{2-\zeta_\epsilon}})} = \gamma_L^{1-\zeta_\epsilon/2}\sqrt{-2\log(1-x_{\zeta_\epsilon,\gamma_L})}$. Therefore, we obtain

$$Q_{1-(1-x_{\zeta_\epsilon,\gamma_L})^{\gamma_L^{2-\zeta_\epsilon}}} \sim \gamma_L^{1-\zeta_\epsilon/2}\sqrt{\frac{2\log(\kappa LN^2)}{\kappa L}}.$$

Combining this result to the fact that $\gamma_L Q_{x_{\zeta_\epsilon,\gamma_L}} \sim \alpha\gamma_L\frac{\log(\kappa LN^2)}{\kappa L}$ we obtain

$$\gamma_L^{-\zeta_\epsilon} \sim \beta\frac{\log(\kappa LN^2)}{\kappa L},$$

where $\beta$ is a positive constant. This yields

$$\mathbb{E}[s_{cr}] \leq \frac{\log(\kappa L N^2)}{\kappa L} + \frac{1}{L} + \frac{\mu}{\kappa L N^2 \log(\kappa L N^2)}(1 + o(1)) + \eta \frac{1}{\kappa^2 L N^2} + \mathcal{O}(\frac{1}{\sqrt{L N^2}})$$

$$= \frac{1}{L}(1 + \frac{\log(\kappa L N^2)}{\kappa}) + \mathcal{O}(\frac{1}{\kappa^2 \sqrt{L N^2}}),$$

where $\kappa = \frac{|\log(\chi)|}{8}$ and $\mu$ is a constant.

2. **Case 2 : Convolutional Neural Networks**
   The proof for CNNs in similar to that of FFNN once we prove that

$$\mathbb{E}\big[\frac{\partial \mathcal{L}}{\partial W_{i,j,\beta}^l}^2\big] = A \, \chi^{L-l}$$

where $A$ is a constant. We have that

$$\frac{\partial \mathcal{L}}{\partial W_{i,j,\beta}^l} = \sum_\alpha \frac{\partial \mathcal{L}}{\partial y_{i,\alpha}^l} \phi(y_{j,\alpha+\beta}^{l-1})$$

and

$$\frac{\partial \mathcal{L}}{\partial y_{i,\alpha}^l} = \sum_{j=1}^n \sum_{\beta \in ker} \frac{\partial \mathcal{L}}{\partial y_{j,\alpha-\beta}^{l+1}} W_{i,j,\beta}^{l+1} \phi'(y_{i,\alpha}^l).$$

Using the approximation of Gradient independence and averaging over the number of channels (using CLT) we have that

$$\mathbb{E}[\frac{\partial \mathcal{L}}{\partial y_{i,\alpha}^l}^2] = \frac{\sigma_w^2 \mathbb{E}[\phi'(\sqrt{q}Z)^2]}{2k+1} \sum_{\beta \in ker} \mathbb{E}[\frac{\partial \mathcal{L}}{\partial y_{i,\alpha-\beta}^{l+1}}^2].$$

Summing over $\alpha$ and using the periodic boundary condition, this yields

$$\sum_\alpha \mathbb{E}[\frac{\partial \mathcal{L}}{\partial y_{i,\alpha}^l}^2] = \chi \sum_\alpha \mathbb{E}[\frac{\partial \mathcal{L}}{\partial y_{i,\alpha}^{l+1}}^2].$$

Here also, on the Ordered phase, the variance $q^l$ and the correlation $c^l$ converge exponentially to their limiting values $q$ and $1$ respectively. As for FFNN, we use the degeneracy approximation that states

- $\forall x \neq x', \alpha, \alpha', c_{\alpha,\alpha'}^l(x, x') \approx 1$,
- $\forall x, q_\alpha^l(x) \approx q$.

Using these approximations, we have

$$\mathbb{E}\big[\frac{\partial \mathcal{L}}{\partial W_{i,j,\beta}^l}^2\big] = \mathbb{E}[\phi(\sqrt{q}Z)^2]\tilde{q}^l(x),$$

where $\tilde{q}^l(x) = \sum_\alpha \mathbb{E}[\frac{\partial \mathcal{L}}{\partial y_{i,\alpha}^l(x)}^2]$ for an input $x$. The choice of $x$ is not important in our approximation.

From the analysis above, we have

$$\tilde{q}^l(x) = \tilde{q}^L(x)\chi^{L-l},$$

so we conclude that

$$\mathbb{E}\big[\frac{\partial \mathcal{L}}{\partial W_{i,j,\beta}^l}^2\big] = A \, \chi^{L-l}$$

where $A = \mathbb{E}[\phi(\sqrt{q}Z)^2]\tilde{q}^L(x)$.

$\square$

After pruning, the network is usually 'deep' in the Ordered phase in the sense that $\chi = f'(1) \ll 1$. To re-place it on the Edge of Chaos, we use the Rescaling Trick.

**Proposition 1** (Rescaling Trick). *Consider a NN of the form (2) or (3) (FFNN or CNN) initialized on the EOC. Then, after pruning, the sparse network is not initialized on the EOC. However, the rescaled sparse network*

$$y^l(x) = \mathcal{F}(\rho^l \circ \delta^l \circ W^l, y^{l-1}(x)) + B^l, \quad for\ l \geq 1, \tag{17}$$

*where*

- $\rho_{ij}^l = \frac{1}{\sqrt{\mathbb{E}[N_{l-1}(W_{i1}^l)^2 \delta_{i1}^l]}}$ *for FFNN of the form (2),*

- $\rho_{i,j,\beta}^l = \frac{1}{\sqrt{\mathbb{E}[n_{l-1}(W_{i,1,\beta}^l)^2 \delta_{i,1,\beta}^l]}}$ *for CNN of the form (3),*

*is initialized on the EOC.*

*Proof.* For two inputs $x, x'$, the forward propagation of the covariance is given by

$$\hat{q}^l(x, x') = \mathbb{E}[y_i^l(x)y_i^l(x')]$$
$$= \mathbb{E}[\sum_{j,k}^{N_{l-1}} W_{ij}^l W_{ik}^l \delta_{ij}^l \delta_{ik}^l \phi(\hat{y}_j^{l-1}(x))\phi(\hat{y}_j^{l-1}(x'))] + \sigma_b^2.$$

We have

$$\frac{\partial \mathcal{L}}{\partial W_{ij}^l} = \frac{1}{|\mathcal{D}|} \sum_{x \in \mathcal{D}} \frac{\partial \mathcal{L}}{\partial y_i^l(x)} \frac{\partial y_i^l(x)}{\partial W_{ij}^l}$$
$$= \frac{1}{|\mathcal{D}|} \sum_{x \in \mathcal{D}} \frac{\partial \mathcal{L}}{\partial y_i^l(x)} \phi(y_j^{l-1}(x)).$$

Under the assumption that the weights used for forward propagation are independent from the weights used for back-propagation, $W_{ij}^l$ and $\frac{\partial \mathcal{L}}{\partial y_i^l(x)}$ are independent for all $x \in \mathcal{D}$. We also have that $W_{ij}^l$ and $\phi(y_j^{l-1}(x))$ are independent for all $x \in \mathcal{D}$. Therefore, $W_{ij}^l$ and $\frac{\partial \mathcal{L}}{\partial W_{ij}^l}$ are independent for all $l, i, j$. This yields

$$\hat{q}^l(x, x') = \sigma_w^2 \alpha_l \mathbb{E}[\phi(\hat{y}_1^{l-1}(x))\phi(\hat{y}_1^{l-1}(x'))] + \sigma_b^2,$$

where $\alpha_l = \mathbb{E}[N_{l-1}(W_{11}^l)^2 \delta_{11}^l]$ (the choice of $i, j$ does not matter because they are iid). Unless we do not prune any weights from the $l^{th}$ layer, we have that $\alpha_l < 1$.
These dynamics are the same as a FFNN with the variance of the weights given by $\hat{\sigma}_w^2 = \sigma_w^2 \alpha_l$. Since the EOC equation is given by $\sigma_w^2 \mathbb{E}[\phi'(\sqrt{q}Z)^2] = 1$, with the new variance, it is clear that $\hat{\sigma}_w^2 \mathbb{E}[\phi'(\sqrt{\hat{q}}Z)^2] \neq 1$ in general. Hence, the network is no longer on the EOC and this could be problematic for training.
With the rescaling, this becomes

$$\hat{q}^l(x, x') = \sigma_w^2 \rho_l^2 \alpha_l \mathbb{E}[\phi(\tilde{y}_1^{l-1}(x))\phi(\tilde{y}_1^{l-1}(x'))] + \sigma_b^2$$
$$= \sigma_w^2 \mathbb{E}[\phi(\tilde{y}_1^{l-1}(x))\phi(\tilde{y}_1^{l-1}(x'))] + \sigma_b^2.$$

Therefore, the new variance after re-scaling is $\tilde{\sigma}_w^2 = \sigma_w^2$, and the limiting variance $\tilde{q} = q$ remains also unchanged since the dynamics are the same. Therefore $\tilde{\sigma}_w^2 \mathbb{E}[\phi'(\sqrt{\tilde{q}}Z)^2] = \sigma_w^2 \mathbb{E}[\phi'(\sqrt{q}Z)^2] = 1$. Thus, the re-scaled network is initialized on the EOC. The proof is similar for CNNs.

$\square$

# D    PROOF FOR SECTION 3 : SBP FOR STABLE RESIDUAL NETWORKS

**Theorem 2** (Resnet is well-conditioned). *Consider a Resnet with either Fully Connected or Convolutional layers and ReLU activation function. Then for all $\sigma_w > 0$, the Resnet is well-conditioned. Moreover, for all $l \in \{1, ..., L\}, m^l = \Theta((1 + \frac{\sigma_w^2}{2})^L)$.*

*Proof.* Let us start with the case of a Resnet with Fully Connected layers. we have that

$$\frac{\partial \mathcal{L}}{\partial W_{ij}^l} = \frac{1}{|\mathcal{D}|} \sum_{x \in \mathcal{D}} \frac{\partial \mathcal{L}}{\partial y_i^l(x)} \frac{\partial y_i^l(x)}{\partial W_{ij}^l}$$

$$= \frac{1}{|\mathcal{D}|} \sum_{x \in \mathcal{D}} \frac{\partial \mathcal{L}}{\partial y_i^l(x)} \phi(y_j^{l-1}(x))$$

and the backpropagation of the gradient is given by the set of equations

$$\frac{\partial \mathcal{L}}{\partial y_i^l} = \frac{\partial \mathcal{L}}{\partial y_i^{l+1}} + \phi'(y_i^l) \sum_{j=1}^{N_{l+1}} \frac{\partial \mathcal{L}}{\partial y_j^{l+1}} W_{ji}^{l+1}.$$

Recall that $q^l(x) = \mathbb{E}[y_i^l(x)^2]$ and $\tilde{q}^l(x, x') = \mathbb{E}[\frac{\partial \mathcal{L}}{\partial y_i^l(x)} \frac{\partial \mathcal{L}}{\partial y_i^l(x')}]$ for some inputs $x, x'$. We have that

$$q^l(x) = \mathbb{E}[y_i^{l-1}(x)^2] + \sigma_w^2 \mathbb{E}[\phi(y_1^{l-1})^2] = (1 + \frac{\sigma_w^2}{2})q^{l-1}(x),$$

and

$$\tilde{q}^l(x, x') = (1 + \sigma_w^2 \mathbb{E}[\phi'(y_i^l(x))\phi'(y_i^l(x'))])\tilde{q}^{l+1}(x, x').$$

We also have

$$\mathbb{E}[\frac{\partial \mathcal{L}}{\partial W_{ij}^l}^2] = \frac{1}{|\mathcal{D}|^2} \sum_{x,x'} t_{x,x'}^l,$$

where $t_{x,x'}^l = \tilde{q}^l(x, x')\sqrt{q^l(x)q^l(x')}f(c^{l-1}(x, x'))$ and $f$ is defined in the preliminary results (Eq 15).

Let $k \in \{1, 2, ..., L\}$ be fixed. We compare the terms $t_{x,x'}^l$ for $l = k$ and $l = L$. The ratio between the two terms is given by (after simplification)

$$\frac{t_{x,x'}^k}{t_{x,x'}^L} = \frac{\prod_{l=k}^{L-1}(1 + \frac{\sigma_w^2}{2}f'(c^l(x, x')))}{(1 + \frac{\sigma_w^2}{2})^{L-k}} \frac{f(c^{k-1}(x, x'))}{f(c^{L-1}(x, x'))}.$$

We have that $f'(c^l(x, x)) = f'(1) = 1$. A Taylor expansion of $f$ near 1 yields $f'(c^l(x, x')) = 1 - l^{-1} + o(l^{-1})$ and $f(c^l(x, x)) = 1 - sl^{-2} + o(l^{-2})$ (see Hayou et al. (2019) for more details).

Therefore, there exist two constants $A, B > 0$ such that $A < \frac{\prod_{l=k}^{L-1}(1 + \frac{\sigma_w^2}{2}f'(c^l(x,x')))}{(1+\frac{\sigma_w^2}{2})^{L-k}} < B$ for all $L$

and $k \in \{1, 2, ..., L\}$. This yields

$$A \leq \frac{\mathbb{E}[\frac{\partial \mathcal{L}}{\partial W_{ij}^l}^2]}{\mathbb{E}[\frac{\partial \mathcal{L}}{\partial W_{ij}^L}^2]} \leq B,$$

which concludes the proof.

For Resnet with convolutional layers, we have

$$\frac{\partial \mathcal{L}}{\partial W_{i,j,\beta}^l} = \frac{1}{|\mathcal{D}|} \sum_{x \in \mathcal{D}} \sum_{\alpha} \frac{\partial \mathcal{L}}{\partial y_{i,\alpha}^l(x)} \phi(y_{j,\alpha+\beta}^{l-1}(x))$$

and

$$\frac{\partial \mathcal{L}}{\partial y_{i,\alpha}^l} = \frac{\partial \mathcal{L}}{\partial y_{i,\alpha}^{l+1}} + \sum_{j=1}^n \sum_{\beta \in ker} \frac{\partial \mathcal{L}}{\partial y_{j,\alpha-\beta}^{l+1}} W_{i,j,\beta}^{l+1} \phi'(y_{i,\alpha}^l).$$

Recall the notation $\tilde{q}_{\alpha,\alpha'}^l(x,x') = \mathbb{E}[\frac{\partial \mathcal{L}}{\partial y_{i,\alpha}^l(x)} \frac{\partial \mathcal{L}}{\partial y_{i,\alpha'}^l(x')}]$. Using the hypothesis of independence of forward and backward weights and averaging over the number of channels (using CLT), we have

$$\tilde{q}_{\alpha,\alpha'}^l(x,x') = \tilde{q}_{\alpha,\alpha'}^{l+1}(x,x') + \frac{\sigma_w^2 f'(c_{\alpha,\alpha'}^l(x,x'))}{2(2k+1)} \sum_\beta \tilde{q}_{\alpha+\beta,\alpha'+\beta}^{l+1}(x,x').$$

Let $K_l = ((\tilde{q}_{\alpha,\alpha+\beta}^l(x,x'))_{\alpha \in [0:N-1]})_{\beta \in [0:N-1]}$ be a vector in $\mathbb{R}^{N^2}$. Writing this previous equation in matrix form, we obtain

$$K_l = (I + \frac{\sigma_w^2 f'(c_{\alpha,\alpha'}^l(x,x'))}{2(2k+1)} U) K_{l+1}$$

and

$$\mathbb{E}[\frac{\partial \mathcal{L}}{\partial W_{i,j,\beta}^l}^2] = \frac{1}{|\mathcal{D}|^2} \sum_{x,x' \in \mathcal{D}} \sum_{\alpha,\alpha'} t_{\alpha,\alpha'}^l(x,x'),$$

where $t_{\alpha,\alpha'}^l(x,x') = \tilde{q}_{\alpha,\alpha'}^l(x,x') \sqrt{q_{\alpha+\beta}^l(x) q_{\alpha'+\beta}^l(x')} f(c_{\alpha+\beta,\alpha'+\beta}^{l-1}(x,x'))$. Since we have $f'(c_{\alpha,\alpha'}^l(x,x')) \to 1$, then by fixing $l$ and letting $L$ goes to infinity, it follows that

$$K_l \sim_{L \to \infty} (1 + \frac{\sigma_w^2}{2})^{L-l} e_1 e_1^T K_L$$

and, from Lemma 2, we know that

$$\sqrt{q_{\alpha+\beta}^l(x) q_{\alpha'+\beta}^l(x')} = (1 + \frac{\sigma_w^2}{2})^{l-1} \sqrt{q_{0,x} q_{0,x'}}.$$

Therefore, for a fixed $k < L$, we have $t_{\alpha,\alpha'}^k(x,x') \sim (1 + \frac{\sigma_w^2}{2})^{L-1} f(c_{\alpha+\beta,\alpha'+\beta}^{k-1}(x,x'))(e_1^T K_L) = \Theta(t_{\alpha,\alpha'}^L(x,x'))$. This concludes the proof.

$\square$

**Proposition 2** (Stable Resnet). *Consider the following Resnet parameterization*

$$y^l(x) = y^{l-1}(x) + \frac{1}{\sqrt{L}} \mathcal{F}(W^l, y^{l-1}), \quad for \ l \geq 2, \tag{18}$$

*then the network is well-conditioned for all choices of $\sigma_w > 0$. Moreover, for all $l \in \{1, ..., L\}$ we have $m^l = \Theta(L^{-1})$.*

*Proof.* The proof is similar to that of Theorem 2 with minor differences. Let us start with the case of a Resnet with fully connected layers, we have

$$\frac{\partial \mathcal{L}}{\partial W_{ij}^l} = \frac{1}{|\mathcal{D}|\sqrt{L}} \sum_{x \in \mathcal{D}} \frac{\partial \mathcal{L}}{\partial y_i^l(x)} \frac{\partial y_i^l(x)}{\partial W_{ij}^l}$$

$$= \frac{1}{|\mathcal{D}|\sqrt{L}} \sum_{x \in \mathcal{D}} \frac{\partial \mathcal{L}}{\partial y_i^l(x)} \phi(y_j^{l-1}(x))$$

and the backpropagation of the gradient is given by

$$\frac{\partial \mathcal{L}}{\partial y_i^l} = \frac{\partial \mathcal{L}}{\partial y_i^{l+1}} + \frac{1}{\sqrt{L}} \phi'(y_i^l) \sum_{j=1}^{N_{l+1}} \frac{\partial \mathcal{L}}{\partial y_j^{l+1}} W_{ji}^{l+1}.$$

Recall that $q^l(x) = \mathbb{E}[y_i^l(x)^2]$ and $\tilde{q}^l(x, x') = \mathbb{E}[\frac{\partial \mathcal{L}}{\partial y_i^l(x)} \frac{\partial \mathcal{L}}{\partial y_i^l(x')}]$ for some inputs $x, x'$. We have

$$q^l(x) = \mathbb{E}[y_i^{l-1}(x)^2] + \frac{\sigma_w^2}{L}\mathbb{E}[\phi(y_1^{l-1}(x))^2] = (1 + \frac{\sigma_w^2}{2L})q^{l-1}(x)$$

and

$$\tilde{q}^l(x, x') = (1 + \frac{\sigma_w^2}{L}\mathbb{E}[\phi'(y_i^l(x))\phi'(y_i^l(x'))])\tilde{q}^{l+1}(x, x').$$

We also have

$$\mathbb{E}[\frac{\partial \mathcal{L}}{\partial W_{ij}^l}^2] = \frac{1}{L|\mathcal{D}|^2}\sum_{x,x'} t_{x,x'}^l,$$

where $t_{x,x'}^l = \tilde{q}^l(x, x')\sqrt{q^l(x)q^l(x')}f(c^{l-1}(x, x'))$ and $f$ is defined in the preliminary results (Eq. 15).

Let $k \in \{1, 2, ..., L\}$ be fixed. We compare the terms $t_{x,x'}^l$ for $l = k$ and $l = L$. The ratio between the two terms is given after simplification by

$$\frac{t_{x,x'}^k}{t_{x,x'}^L} = \frac{\prod_{l=k}^{L-1}(1 + \frac{\sigma_w^2}{2L}f'(c^l(x, x')))}{(1 + \frac{\sigma_w^2}{2L})^{L-k}}\frac{f(c^{k-1}(x, x'))}{f(c^{L-1}(x, x'))}.$$

As in the proof of Theorem 2, we have that $f'(c^l(x, x)) = 1$, $f'(c^l(x, x')) = 1 - l^{-1} + o(l^{-1})$ and $f(c^l(x, x)) = 1 - sl^{-2} + o(l^{-2})$. Therefore, there exist two constants $A, B > 0$ such that $A < \frac{\prod_{l=k}^{L-1}(1 + \frac{\sigma_w^2}{2L}f'(c^l(x, x')))}{(1 + \frac{\sigma_w^2}{2L})^{L-k}} < B$ for all $L$ and $k \in \{1, 2, ..., L\}$. This yields

$$A \leq \frac{\mathbb{E}[\frac{\partial \mathcal{L}}{\partial W_{ij}^l}^2]}{\mathbb{E}[\frac{\partial \mathcal{L}}{\partial W_{ij}^L}^2]} \leq B.$$

Moreover, since $(1 + \frac{\sigma_w^2}{2L})^L \to e^{\sigma_w^2/2}$, then $m^l = \Theta(1)$ for all $l \in \{1, ..., L\}$. This concludes the proof.

For Resnet with convolutional layers, the proof is similar. With the scaling, we have

$$\frac{\partial \mathcal{L}}{\partial W_{i,j,\beta}^l} = \frac{1}{\sqrt{L}|\mathcal{D}|}\sum_{x \in \mathcal{D}}\sum_{\alpha}\frac{\partial \mathcal{L}}{\partial y_{i,\alpha}^l(x)}\phi(y_{j,\alpha+\beta}^{l-1}(x))$$

and

$$\frac{\partial \mathcal{L}}{\partial y_{i,\alpha}^l} = \frac{\partial \mathcal{L}}{\partial y_{i,\alpha}^{l+1}} + \frac{1}{\sqrt{L}}\sum_{j=1}^{n}\sum_{\beta \in ker}\frac{\partial \mathcal{L}}{\partial y_{j,\alpha-\beta}^{l+1}}W_{i,j,\beta}^{l+1}\phi'(y_{i,\alpha}^l).$$

Let $\tilde{q}_{\alpha,\alpha'}^l(x, x') = \mathbb{E}[\frac{\partial \mathcal{L}}{\partial y_{i,\alpha}^l(x)}\frac{\partial \mathcal{L}}{\partial y_{i,\alpha'}^l(x')}]$. Using the hypothesis of independence of forward and backward weights and averaging over the number of channels (using CLT) we have

$$\tilde{q}_{\alpha,\alpha'}^l(x, x') = \tilde{q}_{\alpha,\alpha'}^{l+1}(x, x') + \frac{\sigma_w^2 f'(c_{\alpha,\alpha'}^l(x, x'))}{2(2k+1)L}\sum_{\beta}\tilde{q}_{\alpha+\beta,\alpha'+\beta}^{l+1}(x, x').$$

Let $K_l = ((\tilde{q}_{\alpha,\alpha+\beta}^l(x, x'))_{\alpha \in [0:N-1]})_{\beta \in [0:N-1]}$ is a vector in $\mathbb{R}^{N^2}$. Writing this previous equation in matrix form, we have

$$K_l = (I + \frac{\sigma_w^2 f'(c_{\alpha,\alpha'}^l(x, x'))}{2(2k+1)L}U)K_{l+1},$$

and

$$\mathbb{E}[\frac{\partial \mathcal{L}}{\partial W_{i,j,\beta}^l}^2] = \frac{1}{L|\mathcal{D}|^2}\sum_{x,x' \in \mathcal{D}}\sum_{\alpha,\alpha'}t_{\alpha,\alpha'}^l(x, x'),$$

where $t^l_{\alpha,\alpha'}(x,x') = \tilde{q}^l_{\alpha,\alpha'}(x,x')\sqrt{q^l_{\alpha+\beta}(x)q^l_{\alpha'+\beta}(x')}f(c^{l-1}_{\alpha+\beta,\alpha'+\beta}(x,x'))$. Since we have $f'(c^l_{\alpha,\alpha'}(x,x')) \to 1$, then by fixing $l$ and letting $L$ goes to infinity, we obtain

$$K_l \sim_{L\to\infty} (1+\frac{\sigma_w^2}{2L})^{L-l} e_1 e_1^T K_L$$

and we know from Appendix Lemma 2 (using $\alpha_\beta = \frac{\sigma_w^2}{2L}$ for all $\beta$) that

$$\sqrt{q^l_{\alpha+\beta}(x)q^l_{\alpha'+\beta}(x')} = (1+\frac{\sigma_w^2}{2L})^{l-1}\sqrt{q_{0,x}q_{0,x'}}.$$

Therefore, for a fixed $k < L$, we have $t^k_{\alpha,\alpha'}(x,x') \sim (1+\frac{\sigma_w^2}{2L})^{L-1}f(c^{k-1}_{\alpha+\beta,\alpha'+\beta}(x,x'))(e_1^T K_L) = \Theta(t^L_{\alpha,\alpha'}(x,x'))$ which proves that the stable Resnet is well conditioned. Moreover, since $(1+\frac{\sigma_w^2}{2L})^{L-1} \to e^{\sigma_w^2/2}$, then $m^l = \Theta(L^{-1})$ for all $l$.

$\square$

In the next Lemma, we study the asymptotic behaviour of the variance $q^l_\alpha$. We show that, as $l \to \infty$, a phenomenon of self averaging shows that $q^l_\alpha$ becomes independent of $\alpha$.

**Appendix Lemma 2.** *Let $x \in \mathbb{R}^d$. Assume the sequence $(a_{l,\alpha})_{l,\alpha}$ is given by the recursive formula*

$$a_{l,\alpha} = a_{l-1,\alpha} + \sum_{\beta\in ker} \lambda_\beta a_{l-1,\alpha+\beta}$$

*where $\lambda_\beta > 0$ for all $\beta$. Then, there exists $\zeta > 0$ such that for all $x \in \mathbb{R}^d$ and $\alpha$,*

$$a_{l,\alpha}(x) = (1+\sum_\beta \alpha_\beta)^l a_0 + \mathcal{O}((1+\sum_\beta \alpha_\beta)^l e^{-\zeta l})),$$

*where $a_0$ is a constant and the $\mathcal{O}$ is uniform in $\alpha$.*

*Proof.* Recall that

$$a_{l,\alpha} = a_{l-1,\alpha} + \sum_{\beta\in ker} \lambda_\beta a_{l-1,\alpha+\beta}.$$

We rewrite this expression in a matrix form

$$A_l = U A_{l-1},$$

where $A_l = (a_{l,\alpha})_\alpha$ is a vector in $\mathbb{R}^N$ and $U$ is the is the convolution matrix. As an example, for $k=1$, $U$ given by

$$U = \begin{bmatrix} 1+\lambda_0 & \lambda_1 & 0 & ... & 0 & \lambda_{-1} \\ \lambda_{-1} & 1+\lambda_0 & \lambda_1 & 0 & \ddots & 0 \\ 0 & \lambda_{-1} & 1+\lambda_0 & \lambda_1 & \ddots & 0 \\ 0 & 0 & \lambda_{-1} & 1+\lambda_0 & \ddots & 0 \\ \ddots & \ddots & \ddots & \ddots & & \\ \lambda_1 & 0 & ... & 0 & \lambda_{-1} & 1+\lambda_0 \end{bmatrix}.$$

$U$ is a circulant symmetric matrix with eigenvalues $b_1 > b_2 \geq b_3... \geq b_N$. The largest eigenvalue of $U$ is given by $b_1 = 1 + \sum_\beta \lambda_\beta$ and its equivalent eigenspace is generated by the vector $e_1 = \frac{1}{\sqrt{N}}(1,1,...,1) \in \mathbb{R}^N$. This yields

$$b_1^{-l}U^l = e_1 e_1^T + O(e^{-\zeta l}),$$

where $\zeta = \log(\frac{b_1}{b_2})$. Using this result, we obtain

$$b_1^{-l}A_l = (b_1^{-l}U^l)A_0 = e_1 e_1^T A_0 + O(e^{-\zeta l}).$$

This concludes the proof.

$\square$

Unlike FFNN or CNN, we do not need to rescale the pruned network. The next proposition establishes that a Resnet lives on the EOC in the sense that the correlation between $y_i^l(x)$ and $y_i^l(x')$ converges to 1 at a sub-exponential $\mathcal{O}(l^{-2})$ rate.

**Proposition 3** (Resnet live on the EOC even after pruning). *Let $x \neq x'$ be two inputs. The following statments hold*

1. *For Resnet with Fully Connected layers, let $\hat{c}^l(x, x')$ be the correlation between $\hat{y}_i^l(x)$ and $\hat{y}_i^l(x')$ after pruning the network. Then we have*

$$1 - \hat{c}^l(x, x') \sim \frac{\kappa}{l^2},$$

*where $\kappa > 0$ is a constant.*

2. *For Resnet with Convolutional layers, let $\hat{c}^l(x, x') = \frac{\sum_{\alpha, \alpha'} \mathbb{E}[y_{1,\alpha}^l(x) y_{1,\alpha'}^l(x')]}{\sum_{\alpha, \alpha'} \sqrt{q_\alpha^l(x)} \sqrt{q_{\alpha'}^{l'}(x')}}$ be an 'average' correlation after pruning the network. Then we have*

$$1 - \hat{c}^l(x, x') \gtrsim l^{-2}.$$

*Proof.*    1. Let $x$ and $x'$ be two inputs. The covariance of $\hat{y}_i^l(x)$ and $\hat{y}_i^l(x')$ is given by

$$\hat{q}^l(x, x') = \hat{q}^{l-1}(x, x') + \alpha \mathbb{E}_{(Z_1, Z_2) \sim \mathcal{N}(0, Q^{l-1})}[\phi(Z_1)\phi(Z_2)]$$

where $Q^{l-1} = \begin{bmatrix} \hat{q}^{l-1}(x) & \hat{q}^{l-1}(x, x') \\ \hat{q}^{l-1}(x, x') & \hat{q}^{l-1}(x') \end{bmatrix}$ and $\alpha = \mathbb{E}[N_{l-1} W_{11}^l{}^2 \delta_{11}^l]$.

Consequently, we have $\hat{q}^l(x) = (1 + \frac{\alpha}{2})\hat{q}^{l-1}(x)$. Therefore, we obtain

$$\hat{c}^l(x, x') = \frac{1}{1 + \lambda} \hat{c}^{l-1}(x, x') + \frac{\lambda}{1 + \lambda} f(\hat{c}^{l-1}(x, x')),$$

where $\lambda = \frac{\alpha}{2}$ and $f(x) = 2\mathbb{E}[\phi(Z_1)\phi(xZ_1 + \sqrt{1 - x^2} Z_2)]$ and $Z_1$ and $Z_2$ are iid standard normal variables.

Using the fact that $f$ is increasing (Section B.1), it is easy to see that $\hat{c}^l(x, x') \to 1$. Let $\zeta_l = 1 - \hat{c}^l(x, x')$. Moreover, using a Taylor expansion of $f$ near 1 (Section B.1) $f(x) \underset{x \to 1^-}{=} x + \beta(1 - x)^{3/2} + O((1 - x)^{5/2})$, it follows that

$$\zeta_l = \zeta_{l-1} - \eta \zeta_{l-1}^{3/2} + O(\zeta_{l-1}^{5/2}),$$

where $\eta = \frac{\lambda \beta}{1 + \lambda}$. Now using the asymptotic expansion of $\zeta_l^{-1/2}$ given by

$$\zeta_l^{-1/2} = \zeta_{l-1}^{-1/2} + \frac{\eta}{2} + O(\zeta_{l-1}),$$

this yields $\zeta_l^{-1/2} \underset{l \to \infty}{\sim} \frac{\eta}{2} l$. We conclude that $1 - \hat{c}^l(x, x') \sim \frac{4}{\eta^2 l^2}$.

2. Let $x$ be an input. Recall the forward propagation of a pruned 1D CNN

$$y_{i,\alpha}^l(x) = y_{i,\alpha}^{l-1}(x) + \sum_{j=1}^c \sum_{\beta \in ker} \delta_{i,j,\beta}^l W_{i,j,\beta}^l \phi(y_{j,\alpha+\beta}^{l-1}(x)) + b_i^l.$$

Unlike FFNN, neurons in the same channel are correlated since we use the same filters for all of them. Let $x, x'$ be two inputs and $\alpha, \alpha'$ two nodes in the same channel $i$. Using the Central Limit Theorem in the limit of large $n_l$ (number of channels), we have

$$\mathbb{E}[y_{i,\alpha}^l(x) y_{i,\alpha'}^l(x')] = \mathbb{E}[y_{i,\alpha}^{l-1}(x) y_{i,\alpha'}^{l-1}(x')] + \frac{1}{2k+1} \sum_{\beta \in ker} \alpha_\beta \mathbb{E}[\phi(y_{1,\alpha+\beta}^{l-1}(x))\phi(y_{1,\alpha'+\beta}^{l-1}(x'))],$$

where $\alpha_\beta = \mathbb{E}[\delta_{i,1,\beta}^l W_{i,1,\beta}^l{}^2 n_{l-1}]$.

Let $q^l_\alpha(x) = \mathbb{E}[y^l_{1,\alpha}(x)^2]$. The choice of the channel is not important since for a given $\alpha$, neurons $(y^l_{i,\alpha}(x))_{i\in[c]}$ are iid. Using the previous formula, we have

$$q^l_\alpha(x) = q^{l-1}_\alpha(x) + \frac{1}{2k+1} \sum_{\beta\in ker} \alpha_\beta \mathbb{E}[\phi(y^{l-1}_{1,\alpha+\beta}(x))^2]$$

$$= q^{l-1}_\alpha(x) + \frac{1}{2k+1} \sum_{\beta\in ker} \alpha_\beta \frac{q^{l-1}_{\alpha+\beta}(x)}{2}.$$

Therefore, letting $q^l(x) = \frac{1}{N}\sum_{\alpha\in[N]} q^l_\alpha(x)$ and $\sigma = \frac{\sum_\beta \alpha_\beta}{2k+1}$, we obtain

$$q^l(x) = q^{l-1}(x) + \frac{1}{2k+1} \sum_{\beta\in ker} \alpha_\beta \sum_{\alpha\in[n]} \frac{q^{l-1}_{\alpha+\beta}(x)}{2}$$

$$= (1 + \frac{\sigma}{2})q^{l-1}(x) = (1 + \frac{\sigma}{2})^{l-1}q^1(x),$$

where we have used the periodicity $q^{l-1}_\alpha = q^{l-1}_{\alpha-N} = q^{l-1}_{\alpha+N}$. Moreover, we have $\min_\alpha q^l_\alpha(x) \geq (1 + \frac{\sigma}{2}) \min_\alpha q^{l-1}_\alpha(x) \geq (1 + \frac{\sigma}{2})^{l-1} \min_\alpha q^1_\alpha(x)$.

The convolutional structure makes it hard to analyse the correlation between the values of a neurons for two different inputs. Xiao et al. (2018) studied the correlation between the values of two neurons in the same channel for the same input. Although this could capture the propagation of the input structure (say how different pixels propagate together) inside the network, it does not provide any information on how different structures from different inputs propagate. To resolve this situation, we study the 'average' correlation per channel defined as

$$c^l(x,x') = \frac{\sum_{\alpha,\alpha'} \mathbb{E}[y^l_{1,\alpha}(x)y^l_{1,\alpha'}(x')]}{\sum_{\alpha,\alpha'} \sqrt{q^l_\alpha(x)}\sqrt{q^{l\,\prime}_\alpha(x')}},$$

for any two inputs $x \neq x'$. We also define $\breve{c}^l(x,x')$ by

$$\breve{c}^l(x,x') = \frac{\frac{1}{N^2}\sum_{\alpha,\alpha'} \mathbb{E}[y^l_{1,\alpha}(x)y^l_{1,\alpha'}(x')]}{\sqrt{\frac{1}{N}\sum_\alpha q^l_\alpha(x)}\sqrt{\frac{1}{N}\sum_\alpha q^l_\alpha(x')}}.$$

Using the concavity of the square root function, we have

$$\sqrt{\frac{1}{N}\sum_\alpha q^l_\alpha(x)}\sqrt{\frac{1}{N}\sum_\alpha q^l_\alpha(x')} = \sqrt{\frac{1}{N^2}\sum_{\alpha,\alpha'} q^l_\alpha(x)q^l_\alpha(x')}$$

$$\geq \frac{1}{N^2}\sum_{\alpha,\alpha'} \sqrt{q^l_\alpha(x)}\sqrt{q^l_\alpha(x')}$$

$$\geq \frac{1}{N^2}\sum_{\alpha,\alpha'} |\mathbb{E}[y^l_{1,\alpha}(x)y^l_{1,\alpha'}(x')]|.$$

This yields $\breve{c}^l(x,x') \leq c^l(x,x') \leq 1$. Using Appendix Lemma 2 twice with $a_{l,\alpha} = q^l_\alpha(x)$, $a_{l,\alpha} = q^l_\alpha(x')$, and $\lambda_\beta = \frac{\alpha_\beta}{2(2k+1)}$, there exists $\zeta > 0$ such that

$$c^l(x,x') = \breve{c}^l(x,x')(1 + \mathcal{O}(e^{-\zeta l})). \tag{19}$$

This result shows that the limiting behaviour of $c^l(x,x')$ is equivalent to that of $\breve{c}^l(x,x')$ up to an exponentially small factor. We study hereafter the behaviour of $\breve{c}^l(x,x')$ and use this result to conclude. Recall that

$$\mathbb{E}[y^l_{i,\alpha}(x)y^l_{i,\alpha'}(x')] = \mathbb{E}[y^{l-1}_{i,\alpha}(x)y^{l-1}_{i,\alpha'}(x')] + \frac{1}{2k+1}\sum_{\beta\in ker} \alpha_\beta \mathbb{E}[\phi(y^{l-1}_{1,\alpha+\beta}(x))\phi(y^{l-1}_{1,\alpha'+\beta}(x'))].$$

Therefore,

$$\sum_{\alpha,\alpha'} \mathbb{E}[y_{1,\alpha}^l(x)y_{1,\alpha'}^l(x')]$$

$$= \sum_{\alpha,\alpha'} \mathbb{E}[y_{1,\alpha}^{l-1}(x)y_{1,\alpha'}^{l-1}(x')] + \frac{1}{2k+1}\sum_{\alpha,\alpha'}\sum_{\beta \in ker} \alpha_\beta \mathbb{E}[\phi(y_{1,\alpha+\beta}^{l-1}(x))\phi(y_{1,\alpha'+\beta}^{l-1}(x'))]$$

$$= \sum_{\alpha,\alpha'} \mathbb{E}[y_{1,\alpha}^{l-1}(x)y_{1,\alpha'}^{l-1}(x')] + \sigma \sum_{\alpha,\alpha'} \mathbb{E}[\phi(y_{1,\alpha}^{l-1}(x))\phi(y_{1,\alpha'}^{l-1}(x'))]$$

$$= \sum_{\alpha,\alpha'} \mathbb{E}[y_{1,\alpha}^{l-1}(x)y_{1,\alpha'}^{l-1}(x')] + \frac{\sigma}{2} \sum_{\alpha,\alpha'} \sqrt{q_\alpha^{l-1}(x)}\sqrt{q_\alpha^{l-1\prime}(x')}f(c_{\alpha,\alpha'}^{l-1}(x,x')),$$

where $f$ is the correlation function of ReLU.

Let us first prove that $\breve{c}^l(x,x')$ converges to 1. Using the fact that $f(z) \geq z$ for all $z \in (0,1)$ (Section B.1), we have that

$$\sum_{\alpha,\alpha'} \mathbb{E}[y_{1,\alpha}^l(x)y_{1,\alpha'}^l(x')] \geq \sum_{\alpha,\alpha'} \mathbb{E}[y_{1,\alpha}^{l-1}(x)y_{1,\alpha'}^{l-1}(x')] + \frac{\sigma}{2}\sum_{\alpha,\alpha'}\sqrt{q_\alpha^{l-1}(x)}\sqrt{q_\alpha^{l-1\prime}(x')}c_{\alpha,\alpha'}^{l-1}(x,x')$$

$$= \sum_{\alpha,\alpha'} \mathbb{E}[y_{1,\alpha}^{l-1}(x)y_{1,\alpha'}^{l-1}(x')] + \frac{\sigma}{2}\sum_{\alpha,\alpha'}\mathbb{E}[y_{1,\alpha}^{l-1}(x)y_{1,\alpha'}^{l-1}(x')]$$

$$= (1 + \frac{\sigma}{2})\mathbb{E}[y_{1,\alpha}^{l-1}(x)y_{1,\alpha'}^{l-1}(x')].$$

Combining this result with the fact that $\sum_\alpha q_\alpha^l(x) = (1 + \frac{\sigma}{2})\sum_\alpha q_\alpha^{l-1}(x)$, we have $\breve{c}^l(x,x') \geq \breve{c}^{l-1}(x,x')$. Therefore $\breve{c}^l(x,x')$ is non-decreasing and converges to a limiting point $c$.

Let us prove that $c = 1$. By contradiction, assume the limit $c < 1$. Using equation (19), we have that $\frac{c^l(x,x')}{\breve{c}^l(x,x')}$ converge to 1 as $l$ goes to infinity. This yields $c^l(x,x') \to c$. Therefore, there exists $\alpha_0, \alpha_0'$ and a constant $\delta < 1$ such that for all $l$, $c_{\alpha_0,\alpha_0'}^l(x,x') \leq \delta < 1$. Knowing that $f$ is strongly convex and that $f'(1) = 1$, we have that $f(c_{\alpha_0,\alpha_0'}^l(x,x')) \geq c_{\alpha_0,\alpha_0'}^l(x,x') + f(\delta) - \delta$. Therefore,

$$\breve{c}^l(x,x') \geq \breve{c}^{l-1}(x,x') + \frac{\frac{\sigma}{2}\sqrt{q_{\alpha_0}^{l-1}(x)q_{\alpha_0'}^{l-1}(x')}}{N^2\sqrt{q^l(x)}\sqrt{q^l(x')}}(f(\delta) - \delta)$$

$$\geq \breve{c}^{l-1}(x,x') + \frac{\frac{\sigma}{2}\sqrt{\min_\alpha q_\alpha^1(x)\min_{\alpha'} q_{\alpha'}^1(x')}}{N^2\sqrt{q^1(x)}\sqrt{q^1(x')}}(f(\delta) - \delta).$$

By taking the limit $l \to \infty$, we find that $c \geq c + \frac{\frac{\sigma}{2}\sqrt{\min_\alpha q_\alpha^1(x)\min_{\alpha'} q_{\alpha'}^1(x')}}{N^2\sqrt{q^1(x)}\sqrt{q^1(x')}}(f(\delta) - \delta)$. This cannot be true since $f(\delta) > \delta$. Thus we conclude that $c = 1$.

Now we study the asymptotic convergence rate. From Section B.1, we have that

$$f(x) \underset{x \to 1^-}{=} x + \frac{2\sqrt{2}}{3\pi}(1 - x)^{3/2} + O((1 - x)^{5/2}).$$

Therefore, there exists $\kappa > 0$ such that, close to $1^-$ we have that

$$f(x) \leq x + \kappa(1 - x)^{3/2}.$$

Using this result, we can upper bound $c^l(x,x')$

$$\breve{c}^l(x, x') \leq \breve{c}^{l-1}(x, x') + \kappa \sum_{\alpha, \alpha'} \frac{\frac{1}{N^2} \sqrt{q_\alpha^{l-1}(x)} \sqrt{q_{\alpha'}^{l-1}(x')}}{\sqrt{q^l(x)} \sqrt{q^l(x')}} (1 - c_{\alpha, \alpha'}^l(x, x'))^{3/2}.$$

To get a polynomial convergence rate, we should have an upper bound of the form $\breve{c}^l \leq \breve{c}^{l-1} + \zeta(1 - \breve{c}^{l-1})^{1+\epsilon}$ (see below). However, the function $x^{3/2}$ is convex, so the sum cannot be upper-bounded directly using Jensen's inequality. We use here instead (Pečarić et al., 1992, Theorem 1) which states that for any $x_1, x_2, ... x_n > 0$ and $s > r > 0$, we have

$$\left( \sum_i x_i^s \right)^{1/s} < \left( \sum_i x_i^r \right)^{1/r}. \tag{20}$$

Let $z_{\alpha, \alpha'}^l = \frac{\frac{1}{N^2} \sqrt{q_\alpha^{l-1}(x)} \sqrt{q_{\alpha'}^{l-1}(x')}}{\sqrt{q^l(x)} \sqrt{q^l(x')}}$, we have

$$\sum_{\alpha, \alpha'} z_{\alpha, \alpha'}^l (1 - c_{\alpha, \alpha'}^l(x, x'))^{3/2} \leq \zeta_l \sum_{\alpha, \alpha'} [z_{\alpha, \alpha'}^l (1 - c_{\alpha, \alpha'}^l(x, x'))]^{3/2},$$

where $\zeta_l = \max_{\alpha, \alpha'} \frac{1}{z_{\alpha, \alpha'}^{l\ 1/2}}$. Using the inequality (20) with $s = 3/2$ and $r = 1$, we have

$$\sum_{\alpha, \alpha'} [z_{\alpha, \alpha'}^l (1 - c_{\alpha, \alpha'}^l(x, x'))]^{3/2} \leq \left( \sum_{\alpha, \alpha'} z_{\alpha, \alpha'}^l (1 - c_{\alpha, \alpha'}^l(x, x')) \right)^{3/2}$$

$$= \left( \sum_{\alpha, \alpha'} z_{\alpha, \alpha'}^l - \breve{c}^l(x, x') \right)^{3/2}.$$

Moreover, using the concavity of the square root function, we have $\sum_{\alpha, \alpha'} z_{\alpha, \alpha'}^l \leq 1$. This yields

$$\breve{c}^l(x, x') \leq \breve{c}^{l-1}(x, x') + \zeta(1 - \breve{c}^{l-1}(x, x'))^{3/2},$$

where $\zeta$ is constant. Letting $\gamma_l = 1 - \breve{c}^l(x, x')$, we can conclude using the following inequality (we had an equality in the case of FFNN)

$$\gamma_l \geq \gamma_{l-1} - \zeta \gamma_{l-1}^{3/2}$$

which leads to

$$\gamma_l^{-1/2} \leq \gamma_{l-1}^{-1/2} (1 - \zeta \gamma_{l-1}^{1/2})^{-1/2} = \gamma_{l-1}^{-1/2} + \frac{\zeta}{2} + o(1).$$

Hence we have

$$\gamma_l \gtrsim l^{-2}.$$

Using this result combined with (19) again, we conclude that

$$1 - c^l(x, x') \gtrsim l^{-2}.$$

$\square$

# E    THEORETICAL ANALYSIS OF MAGNITUDE BASED PRUNING (MBP)

In this section, we provide a theoretical analysis of MBP. The two approximations from Appendix A are not used here.

MBP is a data independent pruning algorithm (zero-shot pruning). The mask is given by

$$\delta_i^l = \begin{cases} 1 & \text{if} \quad |W_i^l| \geq t_s, \\ 0 & \text{if} \quad |W_i^l| < t_s, \end{cases}$$

where $t_s$ is a threshold that depends on the sparsity $s$. By defining $k_s = (1-s) \sum_l M_l$, $t_s$ is given by $t_s = |W|^{(k_s)}$ where $|W|^{(k_s)}$ is the $k_s^{\text{th}}$ order statistic of the network weights $(|W_i^l|)_{1 \leq l \leq L, 1 \leq i \leq M_l}$ $(|W|^{(1)} > |W|^{(2)} > ...)$.

With MBP, changing $\sigma_w$ does not impact the distribution of the resulting sparse architecture since it is a common factor for all the weights. However, in the case of different scaling factors $v_l$, the variances $\frac{\sigma_w^2}{v_l}$ used to initialize the weights vary across layers. This gives potentially the erroneous intuition that the layer with the smallest variance will be highly likely fully pruned before others as we increase the sparsity $s$. This is wrong in general since layers with small variances might have more weights compared to other layers. However, we can prove a similar result by considering the limit of large depth with fixed widths.

**Proposition 4** (MBP in the large depth limit). *Assume $N$ is fixed and there exists $l_0 \in [|1, L|]$ such that $\alpha_{l_0} > \alpha_l$ for all $l \neq l_0$. Let $Q_x$ be the $x^{th}$ quantile of $|X|$ where $X \overset{iid}{\sim} \mathcal{N}(0,1)$ and $\gamma = \min_{l \neq l_0} \frac{\alpha_{l_0}}{\alpha_l}$. For $\epsilon \in (0, 2)$, define $x_{\epsilon,\gamma} = \inf\{y \in (0,1) : \forall x > y, \gamma Q_x > Q_{1-(1-x)^{\gamma^{2-\epsilon}}}\}$ and $x_{\epsilon,\gamma} = \infty$ for the null set. Then, for all $\epsilon \in (0,2)$, $x_{\epsilon,\gamma}$ is finite and there exists a constant $\nu > 0$ such that*

$$\mathbb{E}[s_{cr}] \leq \inf_{\epsilon \in (0,2)} \{x_{\epsilon,\gamma} + \frac{\zeta_{l_0} N^2}{1 + \gamma^{2-\epsilon}}(1 - x_{\epsilon,\gamma})^{1+\gamma^{2-\epsilon}}\} + \mathcal{O}(\frac{1}{\sqrt{LN^2}}).$$

Proposition 4 gives an upper bound on $\mathbb{E}[s_{cr}]$ in the large depth limit. The upper bound is easy to approximate numerically. Table 7 compares the theoretical upper bound in Proposition 4 to the empirical value of $\mathbb{E}[s_{cr}]$ over 10 simulations for a FFNN with depth $L = 100$, $N = 100$, $\alpha_1 = \gamma$ and $\alpha_2 = \alpha_3 = \cdots = \alpha_L = 1$. Our experiments reveal that this bound can be tight.

Table 7: Theoretical upper bound of Proposition 4 and empirical observations for a FFNN with $N = 100$ and $L = 100$

| GAMMA | $\gamma = 2$ | $\gamma = 5$ | $\gamma = 10$ |
|---|---|---|---|
| UPPER BOUND | 5.77 | 0.81 | 0.72 |
| EMPIRICAL OBSERVATION | $\approx 1$ | 0.79 | 0.69 |

*Proof.* Let $x \in (0,1)$ and $k_x = (1-x)\Gamma_L N^2$, where $\Gamma_L = \sum_{l \neq l_0} \zeta_l$. We have

$$\mathbb{P}(s_{cr} \leq x) \geq \mathbb{P}(\max_i |W_i^{l_0}| < |W|^{(k_x)}),$$

where $|W|^{(k_x)}$ is the $k_x^{\text{th}}$ order statistic of the sequence $\{|W_i^l|, l \neq l_0, i \in [1 : M_l]\}$; i.e $|W|^{(1)} > |W|^{(2)} > ... > |W|^{(k_x)}$.

Let $(X_i)_{i \in [1:M_{l_0}]}$ and $(Z_i)_{i \in [1:\Gamma_L N^2]}$ be two sequences of iid standard normal variables. It is easy to see that

$$\mathbb{P}(\max_{i,j} |W_{ij}^{l_0}| < |W|^{(k_x)}) \geq \mathbb{P}(\max_i |X_i| < \gamma |Z|^{(k_x)})$$

where $\gamma = \min_{l \neq l_0} \frac{\alpha_{l_0}}{\alpha_l}$.

Moreover, we have the following result from the theory of order statistics, which is a weak version of Theorem 3.1. in Puri and Ralescu (1986)

**Appendix Lemma 3.** *Let $X_1, X_2, ..., X_n$ be iid random variables with a cdf $F$. Assume $F$ is differentiable and let $p \in (0,1)$ and let $Q_p$ be the order $p$ quantile of the distribution $F$, i.e. $F(Q_p) = p$. Then we have*

$$\sqrt{n}(X^{(pn)} - Q_p)F'(Q_p)\sigma_p^{-1} \underset{D}{\to} \mathcal{N}(0,1),$$

*where the convergence is in distribution and $\sigma_p = p(1-p)$.*

Using this result, we obtain

$$\mathbb{P}(\max_i |X_i| < \gamma |Z|^{(k_x)}) = \mathbb{P}(\max_i |X_i| < \gamma Q_x) + \mathcal{O}(\frac{1}{\sqrt{LN^2}}),$$

where $Q_x$ is the $x$ quantile of the folded standard normal distribution.

The next result shows that $x_{\epsilon,\gamma}$ is finite for all $\epsilon \in (0, 2)$.

**Appendix Lemma 4.** *Let* $\gamma > 1$. *For all* $\epsilon \in (0, 2)$, *there exists* $x_\epsilon \in (0, 1)$ *such that, for all* $x > x_\epsilon$, $\gamma Q_x > Q_{1-(1-x)^{\gamma^{2-\epsilon}}}$.

*Proof.* Let $\epsilon > 0$, and recall the asymptotic equivalent of $Q_{1-x}$ given by

$$Q_{1-x} \sim_{x \to 0} \sqrt{-2 \log(x)}$$

Therefore, $\frac{\gamma Q_x}{Q_{1-(1-x)^{\gamma^{2-\epsilon}}}} \sim_{x \to 1} \sqrt{\gamma^\epsilon} > 1$. Hence $x_\epsilon$ exists and is finite. $\square$

Let $\epsilon > 0$. Using Appendix Lemma 4, there exists $x_\epsilon > 0$ such that

$$\mathbb{P}(\max_i |X_i| < \gamma Q_x) \geq \mathbb{P}(\max_i |X_i| < Q_{1-(1-x)^{\gamma^{2-\epsilon}}})$$

$$= (1 - (1-x)^{\gamma^{2-\epsilon}})^{\zeta_{l_0} N^2}$$

$$\geq 1 - \zeta_{l_0} N^2 (1-x)^{\gamma^{2-\epsilon}},$$

where we have used the inequality $(1-t)^z \geq 1 - zt$ for all $(t, z) \in [0, 1] \times (1, \infty)$ and $\beta = \alpha_{l_0} \alpha_{l_0+1}$.

Using the last result, we have

$$\mathbb{P}(s_{cr} \geq x) \leq \beta N^2 (1-x)^{\gamma^{2-\epsilon}} + \mathcal{O}(\frac{1}{\sqrt{LN^2}}).$$

Now we have

$$\mathbb{E}[s_{cr}] = \int_0^1 \mathbb{P}(s_{cr} \geq x) dx$$

$$\leq x_\epsilon + \int_{x_\epsilon}^1 \mathbb{P}(s_{cr} \geq x) dx$$

$$\leq x_\epsilon + \frac{\beta N^2}{1 + \gamma^{2-\epsilon}} (1 - x_\epsilon)^{\gamma^{2-\epsilon}+1} + \mathcal{O}(\frac{1}{\sqrt{LN^2}}).$$

This is true for all $\epsilon \in (0, 2)$, and the additional term $\mathcal{O}(\frac{1}{\sqrt{LN^2}})$ does not depend on $\epsilon$. Therefore there exists a constant $\nu \in \mathbb{R}$ such that for all $\epsilon$

$$\mathbb{E}[s_{cr}] \leq x_\epsilon + \frac{\beta N^2}{1 + \gamma^{2-\epsilon}} (1 - x_\epsilon)^{\gamma^{2-\epsilon}+1} + \frac{\nu}{\sqrt{LN^2}}.$$

We conclude by taking the infimum over $\epsilon$. $\square$

Another interesting aspect of MBP is when the depth is fixed and the width goes to infinity. The next result gives a lower bound on the probability of pruning at least one full layer.

**Proposition 5** (MBP in the large width limit). *Assume there exists* $l_0 \in [1 : L]$ *such that* $\alpha_{l_0} > \alpha_l$ *(i.e.* $v_{l_0} > v_l$) *for all* $l$, *and let* $s_0 = \frac{M_{l_0}}{\sum_l M_l}$. *For some sparsity* $s$, *let* $PR_{l_0}(s)$ *be the event that layer* $l_0$ *is fully pruned before other layers, i.e.*

$$PR_{l_0}(s) = \{|A_{l_0}| = M_{l_0}\} \cap_{l \in [1:L]} \{|A_l| < M_l\},$$

*and let* $PR_{l_0} = \cup_{s \in (s_0, s_{\max})} PR_{l_0}(s)$ *be the event where there exists a sparsity* $s$ *such that layer* $l_0$ *is fully pruned before other layers. Then, we have*

$$\mathbb{P}(PR_{l_0}) \geq 1 - \frac{L\pi^2}{4(\gamma - 1)^2 \log(N)^2} + o(\frac{1}{\log(N)^2}),$$

*where* $\gamma = \min_{k \neq l_0} \frac{\alpha_{l_0}}{\alpha_k}$.

Proposition 5 shows that when the width is not the same for all layers, MBP will result in one layer being fully pruned with a probability that converges to 1 as the width goes to infinity. The larger the ratio $\gamma$ (ratio of widths between the largest and the second largest layers), the faster this probability goes to 1.

The intuition behind Proposition 5 comes from a result in Extreme Value Theory stated in Appendix Lemma 6. Indeed, the problem of pruning one whole layer before the others is essentially a problem of maxima: we prune one whole layer $l_0$ before the others if and only if for all $\max_i |W_i^{l_0}| < \min_{l \neq l_0} \max_i |W_i^l|$. The expected value of $n$ iid standard Gaussian variables is known to scale as $\sqrt{\log n}$ for large $n$; see e.g. Van Handel (2016).

The proof of Proposition 5 relies on the following two auxiliary results.

**Appendix Lemma 5** (Rearrangement inequality (Hardy et al., 1952)). *Let $f, g : \mathbb{R} \to \mathbb{R}^+$ be functions which are either both non-decreasing or non-increasing and let $X$ be a random variable. Then*

$$\mathbb{E}[f(X)g(X)] \geq \mathbb{E}[f(X)]\mathbb{E}[g(X)].$$

**Appendix Lemma 6** (Von Mises (1936)). *Let $(X_i)_{1 \leq i \leq n}$ be iid random variables with common density $f$ and cumulative distribution function $F$. Assume $\lim_{x \to F^{-1}(1)}(\frac{d}{dx}\frac{(1-F(x))}{f(x)}) = 0$, then $\lim_{n \to \infty} \mathbb{P}(\max_i X_i \leq a_n x + b_n) = G(x)$ where $G$ is the Gumbel cumulative distribution function and series $a_n$ and $b_n$ are given by $b_n = F^{-1}(1 - \frac{1}{n})$ and $a_n = \frac{1}{nf(b_n)}$.*

We are now in a position to prove Proposition 5.

*Proof.* Assume there exists $l_0 \in [1 : L]$ such that $\alpha_{l_0} > \alpha_l$ for all $l$. The trick is to see that

$$PR_{l_0} = \{\forall k \neq l_0, \max_i |W_i^{l_0}| < \max_{ij} |W_i^k|\}.$$

Let us prove that

$$\mathbb{P}(PR_{l_0}) \geq \prod_{k \neq l_0} \mathbb{P}(\max_i |W_i^{l_0}| < \max_j |W_i^k|).$$

Let $X = \max_i |W_i^{l_0}|$. We have that

$$\mathbb{P}(PR_{l_0}) = \mathbb{E}[\prod_{k \neq l_0} \mathbb{P}(X < \max_i |W_i^k||X)]$$

using the rearrangement inequality presented in Appendix Lemma 5 with functions $f_i(x) = \mathbb{P}(X < \max_i |W_i^k||X = x)$ which are all non-increasing, we obtain

$$\mathbb{P}(PR_{l_0}) \geq \prod_{k \neq l_0} \mathbb{E}[\mathbb{P}(X < \max_i |W_i^k||X)] = \prod_{k \neq l_0} \mathbb{P}(\max_i |W_i^{l_0}| < \max_i |W_i^k|).$$

In order to deal with the probability $\mathbb{P}(\max_i |W_i^{l_0}| < \max_i |W_i^k|)$, we use Appendix Lemma 6 which is a result from Extreme Value Theory which provides a comprehensive description of the law of $\max_i X_i$ needed in our analysis. In our case, we want to characterise the behaviour of $\max_i |X_i|$ where $X_i$ are iid Gaussian random variables.

Let $\Psi$ and $\psi$ be the cdf and density of a standard Gaussian variable $X$. The cdf of $|X|$ is given by $F = 2\Psi - 1$ and its density is given by $f = 2\psi$ on the positive real line. Thus, $\frac{1-F}{f} = \frac{1-\Psi}{\psi}$ and it is sufficient to verify the conditions of Appendix Lemma 6 for the standard Gaussian distribution. We have $\lim_{x \to F^{-1}(1)} \frac{d}{dx} \frac{1-\Psi(x)}{\psi(x)} = \lim_{x \to F^{-1}(1)} x\frac{(1-\Psi(x))}{\psi(x)} - 1 = x/x - 1 = 0$, where we have used the fact that $x(1 - \Psi(x)) \sim \phi(x)$ in the large $x$ limit.

Let us now find the values of $a_n$ and $b_n$. In the large $x$ limit, we have

$$1 - F(x) = 2\int_x^\infty \frac{e^{-\frac{t^2}{2}}}{\sqrt{2\pi}}dt$$

$$= \sqrt{\frac{\pi}{2}}e^{-\frac{x^2}{2}}(\frac{1}{x} + \frac{1}{x^3} + o(\frac{1}{x^3})).$$

Therefore, one has

$$\log(1 - F(x)) \sim -\frac{x^2}{2}.$$

This yields

$$b_n = F^{-1}(1 - \frac{1}{n}) \sim \sqrt{2 \log n}.$$

Using the same asymptotic expansion of $1 - F(x)$, we can obtain a more precise approximation of $b_n$

$$b_n = \sqrt{2 \log n}\left(1 - \frac{\log(\log n)}{4 \log n} + \frac{\frac{1}{2}\log(\frac{\pi}{4})}{2 \log n} - \frac{\log(\log n)}{8(\log n)^2} + o(\frac{\log(\log n)}{(\log n)^2})\right).$$

Now let us find an approximation for $a_n$. We have

$$\psi(b_n) \sim \frac{\sqrt{2}}{\pi n}\sqrt{\log n}.$$

Therefore, it follows that

$$a_n \sim \frac{\pi}{\sqrt{2 \log n}}.$$

We use these results to lower bound the probability $\mathbb{P}(\max_i |W_i^{l_0}| < \max_i |W_i^k|)$. We have

$$\mathbb{P}(\max_i |W_i^{l_0}| \geq \max_i |W_i^k|) = \mathbb{P}(\max_i |X_i| \geq \gamma_k \max_i |Y_i|),$$

where $\gamma_k = \frac{\alpha_{l_0}}{\alpha_k}$ and $(X_i)$ and $(Y_i)$ are standard Gaussian random variables. Note that $\gamma_k > 1$. Let $A_N = \max_i |X_i|$ and $B_N = \max_i |Y_i|$. We have that

$$\mathbb{P}(A_N \geq \gamma_k B_N) = \mathbb{P}(A_N - \mathbb{E}[A_N] \geq \gamma_k(B_N - \mathbb{E}[B_N]) + \gamma_k \mathbb{E}[B_N] - \mathbb{E}[A_N])$$

$$\leq \mathbb{E}\left[\frac{(A_N - \mathbb{E}[A_N])^2}{(\gamma_k(B_N - \mathbb{E}[B_N]) + \gamma_k \mathbb{E}[B_N] - \mathbb{E}[A_N]))^2}\right] \underset{N \to \infty}{\sim} \frac{\pi^2}{4(\gamma_k - 1)^2 \log(N)^2}.$$

We conclude that for large $N$

$$\mathbb{P}(PR_{l_0}) \geq 1 - \frac{L\pi^2}{4(\gamma - 1)^2 \log(N)^2} + o(\frac{1}{\log(N)^2}),$$

where $\gamma = \min_{k \neq l_0} \frac{\alpha_{l_0}}{\alpha_k}$. $\square$

## F  IMAGENET EXPERIMENTS

To validate our results on large scale datasets, we prune ResNet50 using SNIP, GraSP, SynFlow and our algorithm SBP-SR, and train the pruned network on ImageNet. We train the pruned model for 90 epochs with SGD. The training starts with a learning rate 0.1 and it drops by a factor of 10 at epochs $30, 60, 80$. We report in table 8 Top-1 test accuracy for different sparsities. Our algorithm SBP-SR has a clear advantage over other algorithms. We are currently running extensive simulations on ImageNet to confirm these results.

Table 8: Classification accuracy on ImageNet (Top-1) for ResNet50 with varying sparsities (TODO: These results will be updated to include confidence intervals)

| ALGORITHM | 85% | 90% | 95% |
|---|---|---|---|
| SNIP | 69.05 | 64.25 | 44.90 |
| GRASP | 69.45 | 66.41 | 62.10 |
| SYNFLOW | 69.50 | 66.20 | 62.05 |
| SBP-SR | **69.75** | **67.02** | **62.66** |

Table 9: Test accuracy of pruned neural network on CIFAR10 with different activation functions

| Resnet32 | Algo | 90 | 98 | 99.5 | 99.9 |
|----------|------|-----|-----|------|------|
| Relu | SBP-SR | **92.56(0.06)** | **88.25(0.35)** | **79.54(1.12)** | **51.56(1.12)** |
| | SNIP | 92.24(0.25) | 87.63(0.16) | 77.56(0.36) | 10(0) |
| Tanh | SBP-SR | **90.97(0.2)** | **86.62(0.38)** | **75.04(0.49)** | **51.88(0.56)** |
| | SNIP | 90.69(0.28) | 85.47(0.18) | 10(0) | 10(0) |
| Resnet50 | | | | | |
| Relu | SBP-SR | **92.05(0.06)** | **89.57(0.21)** | **82.68(0.52)** | **58.76(1.82)** |
| | SNIP | 91.64(0.14) | 89.20(0.54) | 80.49(2.41) | 19.98(14.12) |
| Tanh | SBP-SR | **90.43(0.32)** | **88.18(0.10)** | **80.09(0.0.55)** | **58.21(1.61)** |
| | SNIP | 89.55(0.10) | 10(0) | 10(0) | 10(0) |

## G  ADDITIONAL EXPERIMENTS

In Table 10, we present additional experiments with varying Resnet Architectures (Resnet32/50), and sparsities (up to 99.9%) with Relu and Tanh activation functions on Cifar10. We see that overall, using our proposed Stable Resnet performs overall better that standard Resnets.

In addition, we also plot the remaining weights for each layer to get a better understanding on the different pruning strategies and well as understand why some of the Resnets with Tanh activation functions are untrainable. Furthermore, we added additional MNIST experiments with different activation function (ELU, Tanh) and note that our rescaled version allows us to prune significantly more for deeper networks.

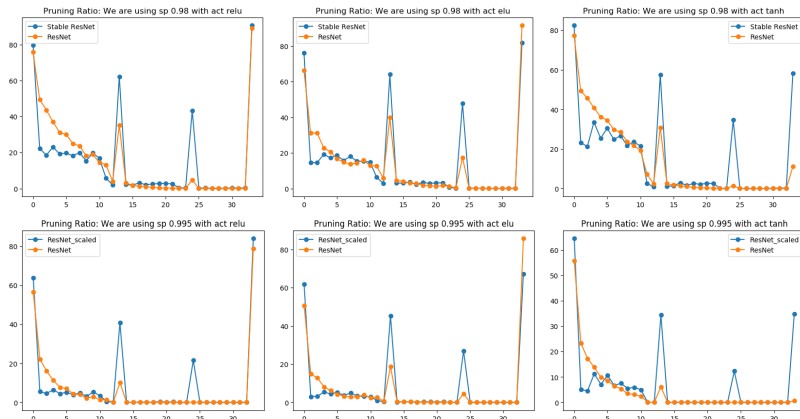

Figure 4: Percentage of pruned weights per layer in a ResNet32 for our scaled ResNet32 and standard Resnet32 with Kaiming initialization

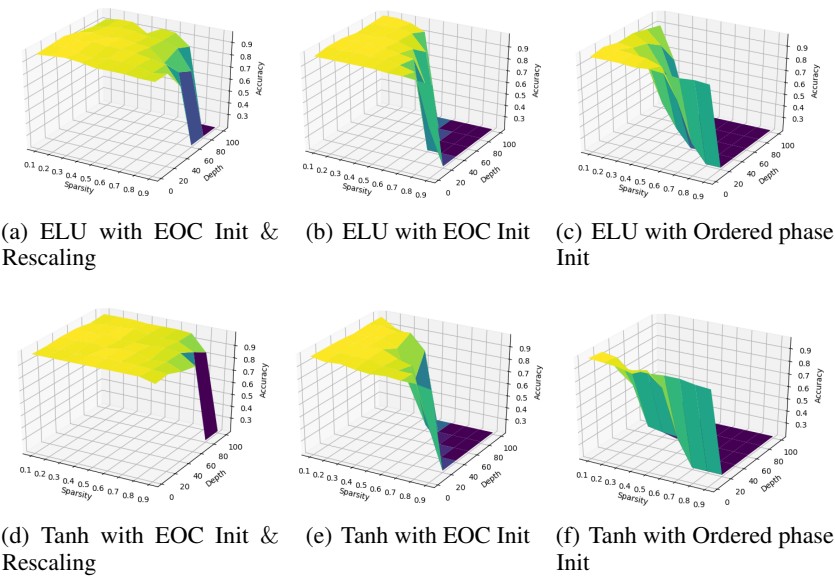

(a) ELU with EOC Init & Rescaling   (b) ELU with EOC Init   (c) ELU with Ordered phase Init

(d) Tanh with EOC Init & Rescaling   (e) Tanh with EOC Init   (f) Tanh with Ordered phase Init

Figure 5: Accuracy on MNIST with different initialization schemes including EOC with rescaling, EOC without rescaling, Ordered phase, with varying depth and sparsity. This figure clearly illustrates the benefits of rescaling very deep and sparse FFNN.

Table 10: Test accuracy of pruned vanilla-CNN on CIFAR10 with different depth/sparsity levels

| Resnet32 | Algo | 90 | 98 | 99.5 | 99.9 |
|---|---|---|---|---|---|
| Relu | SBP-SR | **92.56(0.06)** | **88.25(0.35)** | **79.54(1.12)** | **51.56(1.12)** |
|  | SNIP | 92.24(0.25) | 87.63(0.16) | 77.56(0.36) | 10(0) |
| Tanh | SBP-SR | **90.97(0.2)** | **86.62(0.38)** | **75.04(0.49)** | **51.88(0.56)** |
|  | SNIP | 90.69(0.28) | 85.47(0.18) | 10(0) | 10(0) |
| Resnet50 |  |  |  |  |  |
| Relu | SBP-SR | **92.05(0.06)** | **89.57(0.21)** | **82.68(0.52)** | **58.76(1.82)** |
|  | SNIP | 91.64(0.14) | 89.20(0.54) | 80.49(2.41) | 19.98(14.12) |
| Tanh | SBP-SR | **90.43(0.32)** | **88.18(0.10)** | **80.09(0.0.55)** | **58.21(1.61)** |
|  | SNIP | 89.55(0.10) | 10(0) | 10(0) | 10(0) |

## H  ON THE LOTTERY TICKET HYPOTHESIS

The Lottery Ticket Hypothesis (LTH) (Frankle and Carbin, 2019) states that "randomly initialized networks contain subnetworks that when trained in isolation reach test accuracy comparable to the original network". We have shown so far that pruning a NN initialized on the EOC will output sparse NNs that can be trained after rescaling. Conversely, if we initialize a random NN with any hyperparameters $(\sigma_w, \sigma_b)$, then intuitively, we can prune this network in a way that ensures that the pruned NN is on the EOC. This would theoretically make the sparse architecture trainable. We formalize this intuition as follows.

**Weak Lottery Ticket Hypothesis (WLTH):** *For any randomly initialized network, there exists a subnetwork that is initialized on the Edge of Chaos.*

In the next theorem, we prove that the WLTH is true for FFNN and CNN architectures that are initialized with Gaussian distribution.

**Theorem 3.** *Consider a FFNN or CNN with layers initialized with variances $\sigma_w^2 > 0$ for weights and variance $\sigma_b^2$ for bias. Let $\sigma_{w,EOC}$ be the value of $\sigma_w$ such that $(\sigma_{w,EOC}, \sigma_b) \in EOC$. Then, for all $\sigma_w > \sigma_{w,EOC}$, there exists a subnetwork that is initialized on the EOC. Therefore WLTH is true.*

The idea behind the proof of Theorem 3 is that by removing a fraction of weights from each layer, we are changing the covariance structure in the next layer. By doing so in a precise way, we can find a subnetwork that is initialized on the EOC.

We prove a slightly more general result than the one stated.

**Theorem 4** (Winning Tickets on the Edge of Chaos). *Consider a neural network with layers initialized with variances $\sigma_{w,l} \in \mathbb{R}^+$ for each layer and variance $\sigma_b > 0$ for bias. We define $\sigma_{w,EOC}$ to be the value of $\sigma_w$ such that $(\sigma_{w,EOC}, \sigma_b) \in EOC$. Then, for all sequences $(\sigma_{w,l})_l$ such that $\sigma_{w,l} > \sigma_{w,EOC}$ for all $l$, there exists a distribution of subnetworks initialized on the Edge of Chaos.*

*Proof.* We prove the result for FFNN. The proof for CNN is similar. Let $x, x'$ be two inputs. For all $l$, let $(\delta^l)_{ij}$ be a collection of Bernoulli variables with probability $p_l$. The forward propagation of the covariance is given by

$$\hat{q}^l(x, x') = \mathbb{E}[y_i^l(x) y_i^l(x')]$$
$$= \mathbb{E}[\sum_{j,k}^{N_{l-1}} W_{ij}^l W_{ik}^l \delta_{ij}^l \delta_{ik}^l \phi(\hat{y}_j^{l-1}(x)) \phi(\hat{y}_j^{l-1}(x'))] + \sigma_b^2.$$

This yields

$$\hat{q}^l(x, x') = \sigma_{w,l}^2 p_l \mathbb{E}[\phi(\hat{y}_1^{l-1}(x)) \phi(\hat{y}_1^{l-1}(x'))] + \sigma_b^2.$$

By choosing $p_l = \frac{\sigma_{w,EOC}^2}{\sigma_{w,l}^2}$, this becomes

$$\hat{q}^l(x, x') = \sigma_{w,EOC}^2 \mathbb{E}[\phi(\tilde{y}_1^{l-1}(x)) \phi(\tilde{y}_1^{l-1}(x'))] + \sigma_b^2.$$

Therefore, the new variance after pruning with the Bernoulli mask $\delta$ is $\tilde{\sigma}_w^2 = \sigma_{w,EOC}^2$. Thus, the subnetwork defined by $\delta$ is initialized on the EOC. The distribution of these subnetworks is directly linked to the distribution of $\delta$. We can see this result as layer-wise pruning, i.e. pruning each layer aside. The proof is similar for CNNs. $\square$

Theorem 3 is a special case of the previous result where the variances $\sigma_{w,l}$ are the same for all layers.

# I    ALGORITHM FOR SECTION 2.3

---

**Algorithm 1** Rescaling trick for FFNN

---

**Input:** Pruned network, size $m$
**for** $L = 1$ **to** $L$ **do**
    **for** $i = 1$ **to** $N_l$ **do**
        $\alpha_i^l \leftarrow \sum_{j=1}^{N_{l-1}} (W_{ij})^2 \delta_{ij}^l$
        $\rho_{ij}^l \leftarrow 1/\sqrt{\alpha_i^l}$ **for all** $j$
    **end for**
**end for**

---

