# OpenReview forum: "Robust Pruning at Initialization"
_ICLR.cc/2021/Conference — ICLR 2021 Poster_

### Official Review · AnonReviewer4 · 2020-10-25
**The paper proposed a sensitivity-based pruning at initialization and discussed it from both theoretical and empirical angles.**

**Rating:** 7
**Confidence:** 4

**Review:**

The theoretical analysis is clearly stated in an well-organized way and the derived sparsity bound is reasonable. With FFNN and CNN, a theorem is given to show that the model is trainable only when the initialization on Edge of Chaos (EOC) and also provided a rescaling method to make the pruned NN into EOC regime. With Resnet, it proves the pruning satisfies the EOC condition by default and further provides re-parameterization method to tackle exploding gradients. The experiments well support theoretical results for both FFNN/CNN and resNet.

I would recommend an accept on this paper.

Questions during rebuttal period:
The theorem 2 is stated for resNet with ReLU activation function. Is proposition 2 and section 4.2 only for ReLU as well? Please state the dependence on activation function more clearly in the paper.

One typo: In section 4.1, the figure should be Figure 3 rather than Figure 5.

---

> ### Author Response · Authors · 2020-11-15
> **Response to Reviewer4**
>
> We would like to thank Reviewer 4 for their constructive feedback. Regarding the reviewer’s question “The theorem 2 is stated for resNet with ReLU activation function. Is Proposition 2 and Section 4.2 only for ReLU as well?” Indeed, Theorem 2 and Proposition 2 are only valid for ReLU, and Section 4.2 shows experiments for ResNet with ReLU activation. We will make this clear in the final version.

---

### Official Review · AnonReviewer1 · 2020-10-28
**Rescaling trick to avoid an entire layer pruning phenomena**

**Rating:** 6
**Confidence:** 3

**Review:**

The given paper carries two main contributions: 1) theoretical study of the pruning at initialization (i.e. before training); 2) proposing a new rescaling trick to avoid issues (namely, entire layer pruning) that are common for such pruning mechanisms.

Major concerns:\
  . Theoretical contribution: First of all, I would like to mention that I am not an expert in one-shot pruning and pruning at initialization. However, I strongly believe that the layer pruning problem is commonly observed and studied phenomena (which is stated by authors as well) and it was theoretically studied before. For example, I suggest authors refer to the recent work from [1] where they call it "layer collapse". Furthermore, it was shown that layers with the smaller size have more likelihood of getting entirely pruned. Here, we observe similar behavior (called "layer ill-conditioning") but due to EOC.\
  . Methodological contribution: again, as it is done in [1], the main propose of the rescaling trick is to make sure that the sensitivity score is uniformly distributed to avoid layer pruning. I believe, there are other ways to achieve this goal and thus, contribution is marginal.

[1] Tanaka et al. Pruning neural networks without any data by iteratively conserving synaptic flow. June, 2020.

Major advantageous:\
  . Authors theoretically justify their proposed method;\
  . Experiments show a consistent improvement over the SoA baselines (Snip, Grasp). The improvement is even drastic for ResNet104; However, it will be nice to have a comparison with [1].


---- update after authors' response ----\
Thanks for clarification and providing additional experiments. I'm changing my final evaluation to weak accept. Yes, this paper does provide some interesting insights, but I still think that it has a limited potential impact (see above for major drawbacks).

---

> ### Author Response · Authors · 2020-11-15
> **Response to Reviewer 1**
>
> We thank the reviewer and address their concerns/comments below.
>
> We thank the reviewer for bringing to our attention reference  [1]. Reference [1] indeed addresses similar problems as in our paper such as “layer-collapse” and “layer-ill conditioning”. We came across this paper after having submitted our work to ICLR. However, we would also like to highlight that the first version of our paper (discussing  “layer-collapse” and “layer-ill conditioning’’) appeared on-line several months before [1] (we are happy to share the link with the AC/SAC to prove our claim). The authors of [1] unfortunately did not appear to be aware of our work and do not cite it (even in their 9th November update).
>
> In addition, the reviewer guidelines state:
> “We consider papers contemporaneous if they are published within the last two months. That means, since our full paper deadline is Oct 2, if a paper was published on or after Aug 2, 2020, authors are not required to compare their own work to that paper. Authors are encouraged to cite and discuss all relevant papers, but they may be excused for not knowing about papers not published in peer-reviewed conference proceedings or journals.” Reference [1] is still not published and will only appear in the Proceedings of NeurIPS 2020.
>
> Nevertheless we will be obviously happy to add a comparaison with [1] (currently running experiments) and explain how it differs significantly from our approach:
> - Our paper focuses exclusively on one-shot pruning approaches whereas [1] considers an iterative multi-shot approach. Therefore, we believe it is unfair to compare both methods. We have compared our method to state-of-the-art one-shot pruning approaches.
> - We perform a detailed analysis of a data-dependent gradient based one-shot pruning technique whereas [1] focuses on an iterative data agnostic magnitude based pruning.
> - We take two fundamentally different approaches from [1] to solve the problems of “layer-collapse” and “layer-ill conditioning’’. We consider the problem through the lens of initialisation and gradients, whereas [1] considers conserving the flow responsible for not collapsing the layers after pruning.
>
> Hence, to summarize, our paper and [1] propose drastically different approaches to similar problems under distinct scenarios: one-shot versus multi-shot techniques. We believe both provide valuable insight on what is happening when pruning neural networks. We will provide numerical results that compare our algorithm with [1] as soon as they become available.

---

> ### Author Response · Authors · 2020-11-20
> **Response to Reviewer 1 - Update regarding reference by Tanaka et al.**
>
> The authors of [1] came accross our ICLR reply to your comment which stated "The authors of [1] unfortunately did not appear to be aware of our work and do not cite it (even in their 9th November update)." They kindly sent us an email yesterday entitled "Accidental Miscitation in SynFlow". Here is an extract of their email reproduced below with their agreement (the reference and names in their email have been anonymized):
>
> "We realized we must have miscited your work.  We found that citation number [XXX ] in our paper that we meant to cite your work “XXX et al”, incorrectly refers to a work by “XXX et al.” This was due to my careless bibtex error. We are absolutely aware of and have built on your beautiful work... We have just submitted an update to the arXiv version with the correct citation and will also update the NeurIPS published version."
>
> Note that the title of our paper has been changed for the ICLR submission and their updated manuscript on arXiv today cites the publicly availabe manuscript that has a different title.
>
> [1] Tanaka et al. Pruning neural networks without any data by iteratively conserving synaptic flow. June, 2020.

---

> ### Author Response · Authors · 2020-11-21
> **Update**
>
> ## Update
>
> Regarding the Rev1's comment :
> "Methodological contribution: again, as it is done in [1], the main propose of the rescaling trick is to make sure that the sensitivity score is uniformly distributed to avoid layer pruning. I believe, there are other ways to achieve this goal and thus, contribution is marginal."
>
> As mentioned previously, [1] is actually posterior to our contribution. We respectfully disagree with you about the role of the rescaling trick. The main purpose of the rescaling trick is not to make the sensitivity score uniformly distributed between layers. If it were the case, then indeed there would be other ways to achieve this such as randomly pruning a fraction of the weights in each layer. However, such methods perform poorly (MBP in our paper) compared to other methods. The goal is to design an algorithm that ensures no layer is being fully pruned while outperforming existing methods. Our method suggests a principled one-shot  data-dependent approach to the layer-collapse issue. SynFlow ([1]) is a multi-shot data-agnostic approach, so our approach is very different from that of [1].
> Moreover, **we have compared now our results to SynFlow as you suggested and our method clearly outperforms SynFlow in many scenarios. We refer the reviewer to the revised version of our paper**

---

### Official Review · AnonReviewer2 · 2020-10-28
**A solid paper lacking some experiments**

**Rating:** 6
**Confidence:** 4

**Review:**

### Contents of the paper

The contributions of the paper are three folded: 1) It proposes an essential for pruning at initialization, namely the NN must be initialized with EOC. 2) It proposes a trick to pull the pruned network back into EOC. 3) Some specific research about  ResNet.

### Advantages of the paper

1. The paper seems to be solid with enough motivation and proofs.
2. The experimental results show that the proposed algorithm achieves about 1% higher accuracy on ResNet than other algorithms.

### Weakness

1. Lack of experiments on larger datasets such as ImageNet
2. The authors claim an exploration on FFNN and CNN. However, only results on ResNet are provided. In other words, the effectiveness of the algorithm on other networks are doubtful.
3. Lack of ablation study or case study experiments. For example, what if we prune a CNN without EOC?

#### Updates after rebuttal
1. The authors have provided more results I concerned, which seems to be accord with their conclusions in the paper.
2. Initialization of CNN or other networks is an interesting topic which affects the performance of pruned models. However, there are few papers about the topic. I think the paper is a good example which may arouse more concerns about it.
3. I am willing to increase my rating to 6.

### Question

1. In table 1, why the results on ResNet104 are far better than those on ResNet32 and ResNet50? Especially when the pruning rate is 90%

I'll consider to increase my rating if the authors can provide more convicing results

---

> ### Author Response · Authors · 2020-11-15
> **Response to Reviewer2**
>
> We thank Reviewer 2 for their feedback. We address their comments below.
>
> 1. “Lack of experiments on larger datasets such as ImageNet” : We have illustrated our theoretical results on datasets such as CIFAR100 and TinyImageNet which has 200 classes and they have both shown the benefit of our SBP-SR algorithm over the existing state-of-the-art. It would have certainly been desirable to perform ImageNet experiments. They were not included because of the limited computational resources we had access to at the time of submission.  We are currently running experiments with ImageNet with batch size 16, where we store gradients over iterations  to equate to a batch size of 256. Given our poor computational resources,  this will take a long time to finish  but we will keep updating this response as results are becoming available. Below is an anonymous link to our code for so that anyone can reproduce our results and apply our methodology to challenging problems.
> https://anonymous.4open.science/r/5f3d3740-380c-4fab-ba78-4a271af40e87/
>
> ${\bf Update \hspace{0.1cm}on \hspace{0.1cm}ImageNet}$ :
>
> RESNET50 at 98% sparsity epoch 12
>
> Sparsity 98%  | Top-1 acc | Top-5 acc
>
> SBP-SR     $\hspace{0.9cm}$       | 15.20         | 33.08
>
> GRASP       $\hspace{0.9cm}$      | 8.88           | 19.01
>
>
>
> 2. “The authors claim an exploration on FFNN and CNN. However, only results on ResNet are provided. In other words, the effectiveness of the algorithm on other networks are doubtful.”
> We respectfully  disagree with Reviewer 2 as we have presented results for FFNN in Figure (3). Figure (3)(c) shows that pruning a FFNN on the Ordered phase (i.e. out of the EOC regime) results in quick deterioration as we increase the sparsity. Figure (3)(b) shows that, just by initializing the FFNN on the EOC, we already improve performance and we can prune deeper networks. However, for depth 60 for example, the performance quickly deteriorates as the sparsity level gets over 40%. This is due to the fact that the pruned FFNN is not initialized on the EOC, hence it is not trainable as established by Schoenholz et al. (2017). To resolve this issue, we introduce the Rescaling Trick, which re-puts the pruned network on the EOC and makes it trainable, this is illustrated in Figure (3)(a). Since the theoretical analysis is the same for FFNN and CNN, we only illustrated our results on FFNN. However, a similar behaviour occurs with CNNs. We show in the table below the test accuracy of a vanilla CNN for depths 10, 50, 100 with ReLU on MNIST (3 runs, up to training epoch 80) for sparsity level 50%. As expected, with an initialization out of the EOC (Ordered phase), the pruned CNN is not trained successfully for depths 50 and 100 as some layers are fully pruned.  With an EOC Init, we can train depth 50 but not depth 100, this is due the fact that the pruned CNN is not initialized on the EOC, thus is not trainable (Schoenholz et al. 2017). However, with EOC Init + Rescaling, we can prune and train a CNN of depth 100. We believe this answers the reviewer's question about CNNs, and we are happy to provide further experimental results if the reviewer requests so.
>
> Sparsity 50%                 | Depth 10          | Depth 50            | Depth 100
>
> Ordered phase Init      | 98.12%(0.13)   | 10.15%                | 10.01%
>
> EOC Init                         | 98.20%(0.17)   | 98.75%(0.11)      | 9.98%
>
> EOC Init & Rescaling   | 98.18%(0.21)   | 98.90%(0.07)      | 99.15%(0.08)
>
>
> 3. “Lack of ablation study or case study experiments. For example, what if we prune a CNN without EOC?”
> This is illustrated in Figure (3)(c) for FFNN. We simply cannot prune deeper FFNN if we initialize the network out of the EOC (Ordered phase in Figure (3)(c)). For completeness we have also added results for CNNs  in the above table, which confirms our theory experimentally. We will include these results in the revised paper.
>
> “In table 1, why the results on ResNet104 are far better than those on ResNet32 and ResNet50? Especially when the pruning rate is 90%”
>  The theory of Stable ResNets works in the limit L → infinity. Hence, the algorithm works best practically when the depth is large. Moreover, the pruned ResNet104 has more than twice the number of parameters of pruned ResNet50. We believe that the gap is due to a combination of both facts.
>
> “I'll consider to increase my rating if the authors can provide more convincing results ”
> We will include more FFNN results as well as CNN results in the paper. On ResNets, we provide state-of-the-art results for one-shot pruning techniques on CIFAR100 and TinyImageNet. As discussed above, we are now running simulations on Imagenet and will include them as they are becoming available (keeping in mind we only have access to limited computational resources).

---

> ### Author Response · Authors · 2020-11-21
> **Update**
>
> ## Update
>
> We have uploaded a revised version of the paper, which includes **ImageNet and CNN experiments**. We are currently re-running ImageNet experiments to include standard deviations in the final version of the paper.

---

### Decision · Program_Chairs · 2021-01-07
**Final Decision**

**Decision:**

Accept (Poster)

**Comment:**

The paper proposes a sensitivity-based pruning method at initialization. For fully connection and and convolutional neural networks, it shows that the model is trainable only when the initialization satisfies Edge of Chaos (EOC). The paper also provided a rescaling method so that the pruned network is initialized on the EOC. For Resnet, the paper shows that the proposed pruning satisfies the EOC condition by default and further provides re-parameterization method to tackle exploding gradients. The experiments show the performance of the proposed method on fully connected and convolution neural network, as well as ResNet. There were some concerns about the contribution of the paper compared to that of [1]. I read the two papers carefully and while both papers aim at addressing a similar problem, i.e., pruning at initialization while avoiding layer collapse, the paper provides a different perspective on the problem, and provides enough theoretical contribution and insights to be found helpful and interesting by the community.